

# Estimating precipitation susceptibility in warm marine clouds using multi-sensor aerosol and cloud products from A-Train satellites

Heming Bai[1,2], Cheng Gong[3], Minghuai Wang[1,2], Zhibo Zhang[4], Tristan L'Ecuyer[5]

[1]Institute for Climate and Global Change Research and School of Atmospheric Sciences, Nanjing University, Nanjing, China

[2]Collaborative Innovation Center of Climate Change, Jiangsu Province, China

[3]Institute of Atmospheric Physics, Chinese Academy of Science, Beijing, China

[4]Physics Department, University of Maryland Baltimore County (UMBC), Baltimore, Maryland, USA

[5]Department of Atmospheric and Oceanic Sciences, University of Wisconsin, Madison, Wisconsin, USA

*Corresponding to*: Minghuai Wang (minghuai.wang@nju.edu.cn)

**Abstract.** Precipitation susceptibility to aerosol perturbation plays a key role in understanding aerosol-cloud interactions and constraining aerosol indirect effects. However, large discrepancies exist in the previous satellite estimates of precipitation susceptibility. In this paper, multi-sensor aerosol and cloud products, including those from CALIPSO, CloudSat, MODIS, and AMSR-E from June 2006 to April 2011 are analyzed to estimate precipitation frequency susceptibility $S_{POP}$, precipitation intensity susceptibility $S_I$, and precipitation rate susceptibility $S_R$ in warm marine clouds. We find that $S_{POP}$ strongly depends on atmospheric stability, with stronger reductions in precipitation occurrence observed under more stable environments. Our results show that precipitation susceptibility for drizzle (with -15 dBZ rainfall threshold) is significant different from that for rain (with 0 dBZ rainfall threshold). Onset of drizzle is not as readily suppressed in warm clouds as rainfall while precipitation intensity susceptibility is generally smaller for rain than for drizzle. We find that $S_{POP}$ derived with respect to aerosol index (AI) is about one-third of $S_{POP}$ derived with respect to cloud droplet number concentration (CDNC). Overall, $S_{POP}$ demonstrates relatively robust features throughout independent liquid water path (LWP) products and diverse rain products. In contrast, the behaviors of $S_I$ and $S_R$ are subject to LWP or rain products used to derive them.

## 1 Introduction

Aerosol-cloud interactions play an important role in the climate system and affect the global energy budget and hydrological cycle. Aerosols can change the properties of clouds by two principal pathways: the *cloud albedo effect*, i.e., increasing aerosol loading leads to increased cloud droplet number concentrations (CDNC), which in turn modifies cloud optical properties (Twomey, 1977); and *the cloud lifetime effect*, i.e., aerosol perturbation changes precipitation efficiency, which in turn affects cloud lifetime (Albrecht, 1989). Over the past few decades, numerous methodologies have been developed to understand and quantify the impacts of aerosol-cloud interactions on the climate system. A unique method is to use the so-called "susceptibility" to explain and predict how cloud and precipitation would response if there were some aerosol perturbation. Susceptibility is defined as the derivative of cloud and/or precipitation properties with respect to aerosol related





properties. For example, Platnick and Twomey (1994) proposed a cloud albedo susceptibility as $S_\lambda = \partial A/\partial CDNC$, where A

is cloud albedo and CDNC is cloud droplet number concentration, to quantify the cloud albedo effect of aerosol.

Susceptibility is an inherent property of the aerosol-cloud system. If it can be estimated based on observations, e.g., satellite

remote sensing products, then the impacts of aerosol perturbation can be readily estimated from the product of the

susceptibility and the size of the perturbation, e.g., $\Delta A = S_\lambda \Delta CDNC$.

Precipitation susceptibility has been proposed to evaluate aerosol-cloud-precipitation interactions and to further

constrain cloud lifetime effects in climate models (Feingold and Siebert, 2009; Terai et al., 2012; Wang et al., 2012). It was

first proposed by Feingold and Siebert (2009) and was defined as:

$$S_0 = -\frac{d \ln R}{d \ln CDNC} \tag{1}$$

where R is precipitation intensity (precipitation rate for rainy clouds) and CDNC is cloud droplet number concentration

(Feingold and Siebert, 2009). Sorooshian et al. (2009) further estimated $S_0$ by replacing CDNC with aerosol index (AI).

Wang et al. (2012) proposed an alternative metric, the precipitation frequency susceptibility, defined as:

$$S_{POP} = -\frac{d \ln POP}{d \ln AI} \tag{2}$$

where POP is the probability of precipitation. $S_{POP}$ has been shown to strongly correlate with cloud lifetime effects of

aerosols in global climate models (Wang et al., 2012;Ghan et al., 2016). Terai et al. (2012;2015) further extended the

definition of precipitation susceptibility:

$$S_X = -\frac{d \ln X}{d \ln CDNC} \tag{3}$$

where X can represent precipitation intensity (I, precipitation rate from rainy clouds only), precipitation fraction (POP, or f)

or precipitation rate (R=POP×I, mean precipitation rate from both rainy and non-rainy clouds). Depending on whether I,

POP or R is used in Eq. (3), precipitation intensity susceptibility ($S_I$), precipitation frequency susceptibility ($S_{POP}$ or $S_f$) or

precipitation rate susceptibility ($S_R$) are therefore defined accordingly. Since R can be decomposed into the product of POP

and I, $S_R \approx S_{POP} + S_I$ (Terai et al., 2012, 2015). In addition, some other studies substitute aerosol concentration ($N_A$) or cloud

condensation nuclei (CCN) concentration ($N_{CCN}$) for CDNC to calculate $S_X$ (Terai et al., 2012; Mann et al., 2014).

The behavior and magnitude of aforementioned precipitation susceptibility metrics varies a lot in different studies. For

instance, $S_R$ and $S_{POP}$, using $N_A$ as an aerosol proxy from Terai et al. (2012), both noticeably decrease with increasing LWP,

whereas $S_I$ is flat in the same study. Additionally, previous satellite studies (Wang et al., 2012; Terai et al., 2015; Michibata et

al., 2016) show $S_X$ calculated with respect to CDNC is higher than that with respect to AI. The diverse definitions of

precipitation susceptibility make it challenging to understand susceptibility discrepancies in different studies. An important

objective of this study is to derive these susceptibilities using the same observations in the same context and to better

understand their differences through comparisons.

Another source of uncertainty in the estimation of precipitation susceptibility is the uncertainty associated with the





observation. Among many others, AMSR-E and MODIS are two widely-used satellite cloud retrieval products in aerosol-cloud interaction studies. For instance, Sorooshian et al. (2009) and Wang et al. (2012) both used AMSR-E LWP product to estimate $S_I$ and $S_{POP}$ with respect to AI, respectively. Terai et al. (2015) and Michibata et al. (2016) used MODIS LWP product to estimate $S_I$ with respect to CDNC. Both products have their advantages and limitations, and are both subject to various retrieval uncertainties. AMSR-E has a coarser spatial resolution than MODIS. Its LWP retrievals are available for both daytime and night time, but suffer from instrument noise, cloud detection issues and beam filling effect (Greenwald et al., 2007; Horváth and Gentemann, 2007; Seethala and Horváth, 2010). MODIS LWP retrievals are available only during daytime. The main uncertainty sources in MODIS LWP retrievals include instrument noise, sub-pixel cloud inhomogeneity, three-dimensional radiative effects and uncertainties in ancillary data (Cho et al., 2015; Platnick et al., 2017; Zhang and Platnick, 2011). A recent study by Seethala and Horváth, (2010) revealed several significant differences between ASMR-E and MODIS LWP products, which could contribute to the aforementioned discrepancy of precipitation susceptibility in the literature.

Additionally, different definitions of rain events and/or different methods to derive rain rates could also lead to discrepancy in observation-based estimation of precipitation susceptibility. For example, the rain rate used in Terai et al. (2015) and Michibata et al. (2016) is simply estimated based on a Z-R relationship from CloudSat radar reflectivity profiles measurements. In contrast, Sorooshian et al. (2009) and Wang et al. (2012) used the rain rate reported in CloudSat operational product, which make use not only radar reflectivity but also path-integrated attenuation in the retrieval process (Haynes et al., 2009). The primary satellite data sets used in the previous studies for estimating precipitation susceptibility are listed in Table. 1. To account for the discrepancy in susceptibility as shown in Table. 1, it's important to examine how different LWP and rain data sets affect the estimates of precipitation susceptibility.

Here we estimate precipitation susceptibility using multi-sensor cloud and aerosol products from A-Train satellites. The main objective of this study is to compare precipitation susceptibility estimates based on different retrieval products, and to better understand discrepancies documented in previous studies. As previous studies have shown that aerosol indirect effect and its uncertainties vary in different cloud dynamical regimes (L'Ecuyer et al., 2009; Wang et al., 2012; Zhang et al., 2016), we further examine how precipitation susceptibility might be different under different atmospheric stability conditions. Section 2 introduces different satellite products and methods used to calculate the susceptibility; Section 3 compares precipitation susceptibility estimates from different satellite products and explores how atmospheric stability affects precipitation susceptibility; finally, the discussions are made in Section 4, followed by the summary in Section 5.


## 2 Methods

### 2.1 Satellite datasets

This study mainly uses cloud and aerosol property retrieval products from the Moderate Resolution Imaging Spectroradiometer (MODIS) on Aqua, the Advanced Microwave Scanning Radiometer for Earth Observing System

(AMSR-E) on Aqua, the Cloud Profiling Radar (CPR) on CloudSat and the Cloud-Aerosol Lidar with Orthogonal Polarization (CALIOP) on CALIPSO. All of these satellites operate in the framework of the A-Train constellation (L'Ecuyer and Jiang, 2010; Stephens et al., 2002). Considering most of the warm rainfall occurs in the marine areas (Mülmenstädt et al., 2015) and that satellite retrievals often suffer large uncertainties in the polar regions (Seethala and Horváth, (2010) , the study region is limited to 60 °S to 60 °N over global oceans, covering the period June 2006 to April 2011. Since MODIS

cloud LWP retrieval is only available for daytime, we restrict our analysis to clouds observed in daytime (13:30 local time). MODIS cloud product and CPR radar reflectivity observations used in this study are both provided from the Caltrack datasets, which resample many sensors observations under CALIOP subtrack (see the website of http://www.icare.univ-lille1.fr/projects/calxtract/products for more information). Hence all satellite products have been collocated into the CALIOP tracks with the horizontal resolution of 5 km. The main satellite datasets used in this study are

briefly listed in Table. 2.

### 2.1.1 AI and CDNC

Three aerosol products are used in the study: MODIS Level 3 daily mean atmosphere product (MYD08_D3, Collection 6), MODIS Level 2 aerosol product (MYD04_L2, Collection 6) and CALIOP Level 2 aerosol layer product (CAL_LID_L2_05kmALay, Version 3.01). The one degree daily mean product of MYD08_D3 is aggregated from

MYD04_L2 with 10 km horizontal resolution (Hubanks et al., 2016). Horizontal resolution of column aerosol optical depth from CAL_LID_L2_05kmALay product is 5 km. Aerosol property in this dataset is obtained by averaging the 16 aerosol extinction profiles with 333 m of native resolution along track (Young and Vaughan, 2009).

Since AI is a better proxy for CCN concentrations as compared to AOD (Nakajima et al., 2001), AI is calculated as one of the proxy for CCN based on the definition of AI=AOD×AE, where AOD and AE are aerosol optical depth and Ångström

coefficient, respectively. For MODIS, AOD at 0.55 μm reported from MYD08_D3 and MYD04_L2 products are based on the Dark Target algorithm over ocean (Kaufman et al., 1997; Tanré et al., 1997; Levy et al., 2013). For CALIOP, AOD at wavelength of 0.532 μm is obtained from the CAL_LID_L2_05kmALay product (Vaughan et al., 2004). Unlike MODIS AE, which is directly reported in aerosol products, AE measurement for CALIOP is calculated based on AOD at 1.064μm and 0.532μm from CAL_LID_L2_05kmALay product (Bréon et al., 2011). Our data screening for CAL_LID_L2_05kmALay

follows a previous study by Kim et al. (2013).

Three aerosol products used in this study are listed in the Table 2. It should be noted that all aerosol samples are under



cloud free conditions and are selected in close proximity to clouds pixels. Retrievals of aerosol properties from passive sensors and lidar observation are both affected by clouds near the aerosol, and thereby result in overestimation for aerosol property (Chand et al., 2012; Tackett and Di Girolamo, 2009). The extent of this overestimation may be different among different sensors. This effect, however, would likely impact all metrics in a similar way and we would not expect this effect

would impact qualitative comparisons between different metrics.

CDNC is derived from the cloud optical thickness $\tau$ and cloud top effective radius $r_{eff}$, both reported in the MODIS level 2 cloud product (namely, MYD06_L2), based on the following formula (Bennartz, 2007; Quaas et al., 2006; Wood and Hartmann, 2006):

$$CDNC = \alpha \, \tau^{0.5} r_{eff}^{-2.5} \tag{4}$$

where the coefficient $\alpha = 1.37 \times 10^{-5} m^{-0.5}$ is estimated based on the assumption that cloud vertical structure follows the classic adiabatic growth model (Quaas et al., 2006). To reduce the uncertainty when deriving CDNC in clouds, pixels where the cloud optical depth is less than 3 and cloud fraction is less than 100% are excluded (Cho et al., 2015; Zhang and Platnick, 2011). Additionally, we limit our analysis to warm clouds by screening cloud pixels with cloud top temperature warmer than 273K. Under these screening criteria, Kubar et al. (2009) found that most (93%) of warm clouds are single layered.

Therefore, our analysis mainly focuses on single-layer clouds.

### 2.1.2 LWP

Cloud LWP for MODIS is diagnosed from solar reflectance observations of $r_{eff}$ and $\tau$ as (Platnick et al., 2003):

$$LWP = a\rho_w \tau r_{eff} \tag{5}$$

where $\rho_w$ denotes the liquid water density and $a$ is a constant determined by the assumed vertical variation in cloud droplet

size (Greenwald, 2009). For a vertically homogeneous cloud, $a = 2/3$ (Bennartz, 2007), and $a = 5/9$ when the adiabatic assumption is applied (Szczodrak et al., 2001; Wood and Hartmann, 2006). A recent study by (Miller et al., 2016) provides a systematic investigation of the impacts of cloud vertical structure on MODIS LWP retrievals. To be consistent with the adiabatic assumption used in Eq. (4) for estimating CDNC, $a = 5/9$ is applied here.

The other LWP retrieval comes from AMSR-E Level 2B Global Swath Ocean Product (Wentz and Meissner, 2004).

Unlike retrieving from solar reflectance of visible near-infrared (VNIR) for MODIS, LWP for AMSR-E is directly derived from brightness temperatures based on liquid-sensitive 37 GHz channel measurements (Seethala and Horváth, 2010). More information of retrieval technique of AMSR-E LWP is documented in Wentz and Meissner (2000). Horizontal resolution of AMSR-E LWP product (12km) is also different from MODIS LWP product (5km).

### 2.1.3 Precipitation

Precipitation datasets used in this study are derived from three different products from the CloudSat CPR, namely



2B-GEOPROF, 2C-PRECIP-COLUMN and 2C-RAIN-PROFILE. All the estimates are limited to cloudy profiles by using 2B-GEOPROF cloud mask, which is set to greater than 20 (King et al., 2015). For the 2B-GEOPROF product (Marchand et al., 2008), the maximum radar reflectivity for each cloudy profile is used to define rain event and to estimate rain rate. More specifically, rain rate is obtained by employing the reflectivity-rainfall (Z-R) relationship at cloud base ($Z=25R^{1.3}$ from

Comstock et al., 2004), and a radar reflectivity threshold is used to distinguish between drizzling and nondrizzling clouds (Terai et al., 2012, 2015).

The empirical Z-R relationship, however, does not account for multiple-scattering by raindrops and attenuation due to both gases and hydrometeors, which poses major challenges for calculation of rain rate, especially surface rain rate (Lebsock and L'Ecuyer, 2011). To address those challenges, Haynes et al. (2009) introduced a full rainfall retrieval algorithm, which is

the basis of the 2C-PRECIP-COLUMN product. The algorithm first makes use of path-integrated attenuation (PIA) derived from measurements of radar backscatter over ocean surface in conjunction with surface wind speed and sea surface temperature. Surface rain rate is then estimated based on a simple algorithm using the PIA. For the 2C-PRECIP-COLUMN product, rain event is identified by using rain likelihood mask. Here, we use flag of "rain certain" to define rain event, which means attenuation-corrected reflectivity near surface is above 0dBZ (Haynes et al., 2009).

2C-PRECIP-COLUMN assumes a constant vertical rain profile in the precipitating column (Haynes et al., 2009), which may not be suitable for warm rain where vertical variation of rain profile is significant (Lebsock and L'Ecuyer, 2011).   To address this issue, CloudSat developed a third rain product, 2C-RAIN-PROFILE that utilizes the complete vertically-resolved reflectivity profile observed by the CPR and incorporates a subcloud evaporation model. 2C-RAIN-PROFILE also uses MODIS cloud visible properties to constrain cloud water in its retrieval algorithm (Lebsock

and L'Ecuyer, 2011). Note that the 2C-RAIN-PROFILE algorithm directly uses the precipitation occurrence flag from 2C-PRECIP-COLUMN, to define rain events. Thus the probability of precipitation (POP) is the same for both rain products. Note that surface rain rates are only retrieved for those pixels that identified as rain certain in 2C-RAIN-PROFILE product (Lebsock and L'Ecuyer, 2011). Overall, three rain rate datasets in this study are significant different: rain rate directly estimated from 2B-GEOPROF represents the maximum rainfall rate, precipitation from 2C-PRECIP-COLUMN is the

column-mean rainfall rate, and rain rate from 2C-RAIN-PROFILE stands for surface rainfall rate.

### 2.2 Meteorological datasets

Aerosol-cloud-precipitation interactions and precipitation susceptibility have been shown to depend on cloud regimes (L'Ecuyer et al., 2009). Following Klein and Hartmann. (1993), we use the lower-tropospheric static stability (LTSS), which is defined as the difference in potential temperature between 700hPa and the surface, to separate different cloud regimes. In

this study, unstable and stable environments are defined as LTSS less than 13.5K and LTSS larger than 18K, respectively. Pixels where LTSS between 13.5K and 18K are defined as the mid-stable environment (Wang et al., 2012). The European



Centre for Medium-Range Weather Forecasts Auxiliary (ECMWF-AUX) product, as an ancillary CloudSat product that contains temperature and pressure within each CPR bin, is used to calculate LTSS.

**2.3 Precipitation susceptibility calculation**

Following previous studies (Feingold and Siebert, 2009; Sorooshian et al., 2009; Wang et al., 2012; Terai et al., 2012,

2015), precipitation susceptibility is generally defined as:

$$S_{X\_Y} = -\frac{d \ln X}{d \ln Y} \qquad (6)$$

where X can be substituted by POP (precipitation frequency), I (precipitation intensity), or R (R=POP×I, precipitation rate), and Y indicates AI or CDNC. Consequently, six different precipitation susceptibilities can be obtained from the observations described above. To constrain cloud macrophysical environment, all samples are sorted according to their LWP values first

and then divided into 10 LWP bins. The ratio of the number of pixels in each bin to the total pixels ranges from 5% to 14%. For each LWP bin, samples are sorted by AI or CDNC, and ten AI/CDNC bins are equally divided to calculate POP, mean I, R, AI and CDNC within each AI/CDNC bin. Finally, the values of $S_{X\_Y}$ are derived by linear regression in log-log space.

**3 Results**

**3.1 $S_{X\_AI}$ versus $S_{X\_CDNC}$**

$S_{X\_AI}$ and $S_{X\_CDNC}$ as a function of LWP are shown in Fig. 1. Here LWP from MODIS and rain data from 2B-GEOPROF with a rain threshold of -15dBZ are used, to better compare with other satellite studies (Terai et al., 2015; Michibata et al., 2016). Here AI is estimated by using MYD04 dataset and detailed comparison among different aerosol products will be discussed in Section 3.2.

Consistent with previous studies, $S_{X\_AI}$ are generally much smaller than $S_{X\_CDNC}$ as shown in Fig. 1. $S_{POP\_AI}$ from Wang

et al. (2012) is less than 0.2 over all LWP bins, while Terai et al. (2015) showed that $S_{POP\_CDNC}$ decreases with increasing LWP, ranging from 1 to 0, and $S_{R\_CDNC}$ is maintained at around 0.5. Fig. 1b further shows $S_{I\_CDNC}$ monotonically increases with LWP, followed by a slight decrease. The $S_{I\_CDNC}$ peak, around 0.6 with LWP 400 $gm^{-2}$, is in good agreement with result of Michibata et al. (2016), which suggested that the turning point corresponds to conversion process shifting from the autoconversion to accretion regime.

To account for discrepancy between $S_{X\_AI}$ and $S_{X\_CDNC}$, we use the condition probability method (Gryspeerdt et al., 2016) to explore relationships between AI and CDNC. As Fig. 2a shows, the majority of CDNC values concentrate on the intervals between 20 $cm^{-3}$ to 100 $cm^{-3}$, representing an upward tendency with increasing AI over global oceans. The similar feature of CDNC with AI is also shown in different LTSS conditions (Fig. 2b, 2c and 2d). Note that fluctuation of the curve at high AI results from the small number of effective pixels, especially in unstable condition.





To formally account for the relationship between CDNC and AI, $S_{X\_AI}$ can be decomposed into two parts:

$$S_{X\_AI} = -\frac{d \ln X}{d \ln AI} = -\frac{d \ln X}{d \ln CDNC}\frac{d \ln CDNC}{d \ln AI} = S_{X\_CDNC}\frac{d \ln CDNC}{d \ln AI} \qquad (7)$$

where dlnCDNC/dlnAI is the link between $S_{X\_AI}$ and $S_{X\_CDNC}$. $S_{X\_AI}$ is expected to be smaller than $S_{X\_CDNC}$ if dlnCDNC/dlnAI is smaller than 1. Fig. 3 shows dlnCDNC/dlnAI over global oceans, which is calculated by log-log linear

regressions in each MODIS LWP bin. dlnCDNC/dlnAI is smaller than 0.4, which explains why $S_{X\_AI}$ is generally smaller than $S_{X\_CDNC}$. Table 3 further shows the LWP-weighted mean of dlnCDNC/dlnAI, $S_{X\_AI}$, and $S_{X\_CDNC}$ over global oceans. Our results are consistent with the previous satellite observations. For instance, $S_{POP\_AI}$ is equal to 0.11 in our results obtained from AMSR-E LWP, close to the value of 0.12 in Wang et al. (2012), and our $S_{R\_CDNC}$ derived from MODIS LWP is 0.74, similar to that (0.6) in Terai et al. (2015). Since the global mean dlnCDNC/dlnAI is about 0.3, we would expect $S_{X\_AI}$ is

about one-third of $S_{X\_CDNC}$, according to Eq. (7). Table 3 shows that this relationship is generally true for $S_{POP}$, but less so for $S_I$, especially for $S_I$ calculated based on MODIS LWP.

Table 3 further demonstrates that $S_R \approx S_I + S_{POP}$ is generally true for different LWP products and over different stability regimes, consistent with Terai et al. (2015).

### 3.2 $S_{X\_AI}$ from different aerosol products

Now we explore how precipitation susceptibility estimates might be different from different aerosol products (i.e., MYD04, MYD08 and CAL_LID_L2_05kmALay). As shown in Figure 4, despite differences in their horizontal resolutions (10 km versus 1 degree), $S_{X\_AI}$ calculated from MYD04 and MYD08 agrees well (Fig. 4a and Fig. 4b), which may result from the fact that aerosol layers are likely homogeneous over relatively large spatial scales less than 200 km (Anderson et al., 2003), especially over global oceans. In addition, McComiskey and Feingold (2012) found that the statistics (i.e., min, max

and variance) of AOD are constant between MYD04 and MYD08 products over the northeast Pacific Ocean for a given day. Although not shown here, the probability distributions of AI derived from MYD04 and MYD08 products are qualitatively similar over global oceans. In comparison with the results based on MODIS retrievals, $S_{X\_AI}$ obtained from CALIOP (Fig. 4c) is smaller and relatively flat across all LWP bins. Further test shows that $S_{X\_AI}$ using CALIOP AOD but MYD04 AE agrees better with that based on MODIS aerosol products (Fig. 4d). This suggests that differences in AE estimates from MODIS and

CALIOP largely explain the discrepancy between two aerosol products. Previous studies indicate that MODIS and CALIOP AOD are poorly correlated (e.g., Costantino and Bréon, 2010; Kim et al., 2013; Kittaka et al., 2011; Ma et al., 2013). Our results suggest that differences in AOD retrievals can lead to differences in AE estimates and further affect AI and precipitation susceptibly estimates. For the rest of the paper, AI from MYD04 is used, unless otherwise stated.

### 3.3 $S_{X\_Y}$ from different LWP dataset

Figure 5 shows the behavior of $S_{POP}$ and $S_I$ based on different LWP data sets (i.e., AMSR-E and MODIS LWP).




Estimates of rain rate and rain events are based on 2B-GEOPROF with -15dBZ threshold as mentioned in Section 2.1.3. Here we focus on characteristics of $S_{POP}$ and $S_I$ since $S_R \approx S_I + S_{POP}$ as mentioned in Section 3.1. As shown in Fig. 5a, $S_{POP\_CDNC}$ based on MODIS LWP is similar to that calculated based on AMSR-E LWP. This consistency is also found for $S_{POP\_AI}$. In contrast, $S_{I\_CDNC}$ and $S_{I\_AI}$ calculated based on two LWP products are quite different (Fig. 5b). $S_{I\_CDNC}$ based on

MODIS LWP are significantly larger than that based on AMSR-E LWP over all LWP bins (see squares in Fig. 5b), while $S_{I\_AI}$ from two LWP products shows an opposite pattern: $S_{I\_AI}$ based on MODIS LWP is lower than that based on AMSR-E LWP (see points in Fig. 5b). These features of discrepancies in $S_I$ between MODIS and AMSR-E LWP are still applicable to $S_{POP}$, though the magnitude is much smaller (Fig. 5a).

Fig. 5b shows that LWP value where $S_{I\_CDNC}$ peaks based on MODIS LWP is larger than that based on AMSR-E LWP.

Large eddy simulation analysis by Duong et al. (2011) showed a similar shift in LWP with changing spatial resolutions, which is attributed to reduction in mean LWP at coarser resolutions. However, Fig. 6 shows that there is no systematic shift in the frequency distribution of LWP between two LWP products, regardless of precipitation or non-precipitation samples.

To better understand the discrepancy in precipitation susceptibility estimates from two LWP products in Fig. 5, we plot POP and intensity as a function of CDNC/AI in log space for each LWP bin obtained from MODIS and AMSR-E. Fig. 7a-7d

shows that the relationships between POP and CDNC (AI) from MODIS LWP are similar to that from AMSR-E LWP. In contrast, intensity versus CDNC (AI) between two LWP products shows significant differences (Fig. 7e-7h). Fig. 7f shows intensity is positively correlated with CDNC at low CDNC for high AMSR-E LWP bins, which helps to explain why $S_{I\_CDNC}$ from AMSR-E LWP is smaller than that from MODIS, especially at high LWP bins (Fig. 5b).

Combining Eq. (4) and (5), CDNC from MODIS can be reformulated as a function of LWP and $r_{eff}$:

$$CDNC = \alpha (a\rho_w)^{-0.5} LWP^{0.5} r_{eff}^{-3} \qquad (8)$$

where $\alpha$, $a$ and $\rho_w$ are all constant. Accordingly, $r_{eff}$ decreases with increasing CDNC for any given MODIS LWP bin, and larger CDNC leads to smaller $r_{eff}$, which further results in reduction in precipitation efficiency, as shown in Fig. 7e. The CDNC-$r_{eff}$ relationship still holds when data is binned by AMSR-E LWP and $r_{eff}$ decreases with increasing CDNC even at larger LWP AMSRE-LWP bins (Fig. 8a). We would then expect rain intensity still decreases with increasing CDNC for the

AMSR-E LWP at low CDNC. So then what might lead to increases in precipitation intensity with increasing CDNC at low CDNC when data is binned according to constant AMSR-E LWP (Fig. 7f)? Our analysis suggests that this might come from the discrepancies in two LWP products under low CDNC. Fig. 8b shows that, under constant AMSR-E LWP, MODIS LWP significantly varies with CDNC (Fig. 8b). In particular, MODIS LWP rapidly increases with CDNC at low CDNC, which might help to explain why rain intensity increases with increasing CDNC at low CDNC under constant AMSR-E LWP,

which further leads to much smaller $S_{I\_CDNC}$ from AMSR-E LWP. Our results further indicate that rain intensity retrieval from CloudSat might be more consistent with LWP retrieval from MODIS than that from AMSR-E, as under constant





AMSR-E LWP, rain intensity increases with increasing MODIS LWP at low CDNC (Fig. 7f and Fig. 8b).

It is interesting to note that, for rainy pixels, difference in LWP between MODIS and AMSR-E varies with CDNC. Under constant AMSR-E LWP (larger than 200 g m$^{-2}$), MODIS LWP dramatically increases with increasing CDNC at lower CDNC (< ~25 cm$^{-3}$). These features are also applicable to non-rainy samples (not shown). Further studies are needed to understand the aforementioned discrepancy.

### 3.4 $S_{X\_Y}$ from different rainfall definition

Given that rainy samples may be dominated by different precipitation process (e.g., autoconversion vs. accretion process) with increasing threshold for defining a rainfall event (Jung et al., 2016), precipitation susceptibility may be changed when we apply different rainfall thresholds. To examine this, we plot $S_{POP}$ and $S_I$ under different thresholds (i.e., -15dBZ and 0dBZ of maximum radar reflectivity) used to define a rain event based on 2B-GEOPROF products. These thresholds of -15dBZ and 0dBZ correspond to approximately 0.14 and 2 mm d$^{-1}$, respectively (Comstock et al., 2004). Hence, precipitation susceptibilities under these two thresholds can be referred to as drizzle (>-15dBZ) and rain (>0dBZ) susceptibilities. As Fig. 9 indicates, difference in $S_{X\_AI}$ between drizzle and rain is, at first glance, less evident compared to $S_{X\_CDNC}$. This can be partly attributed to the low values of $S_{X\_AI}$ themselves. Relative differences in $S_{X\_AI}$ are even larger than that of $S_{X\_CDNC}$ at low AMSR-E LWP (not shown). Fig. 9a and Fig. 9c show that rain $S_{POP\_AI}$ is higher than that of drizzle over most LWP bins, which is consistent with results from Wang et al. (2012).

Rainfall definition significantly impacts $S_{POP\_CDNC}$ and $S_{I\_CDNC}$: increasing the threshold results in reduction of $S_{I\_CDNC}$ over all LWP and, by contrast, leads to distinctly increase in $S_{POP\_CDNC}$, especially at moderate LWP (see Fig. 9). These overall changes in $S_{I\_CDNC}$ and $S_{POP\_CDNC}$ after increasing the threshold are consistent with Terai et al. (2015). The observational result from Mann et al. (2014) have also shown an evident increase in $S_{POP\_CDNC}$ at high LWP with increasing thresholds. The systematic increase in $S_{POP\_CDNC}$ may result from larger proportion of non-drizzling samples with increasing threshold. The reduction of $S_{I\_CDNC}$ is in agreement with previous studies (Duong et al., 2011; Jung et al., 2016). Although not shown here, for a fixed threshold, there is no significant discrepancy between the results of $S_{I\_CDNC}$ and $S_{R\_CDNC}$ based on different Z-R relationships (Z=25R$^{1.3}$ and Z=302R$^{0.9}$ are used both from Comstock et al. 2004, which aim at cloud base and surface rain rate, respectively), which is consistent with the result from Terai et al. (2012).

Overall, our results show that $S_{POP}$ and $S_I$ are both sensitive to the rainfall definition and that $S_{POP}$ is greater for rain while $S_I$ is greater for drizzle. Our results further imply that onset of drizzle is not as readily suppressed in warm clouds as rainfall (i.e., $S_{POP}$ is greater for rain than for drizzle).

While the response of precipitation susceptibility to change in threshold shows the same pattern between MODIS and AMSR-E LWP (Fig. 9), the extent to which susceptibility changes with increasing threshold is quite different between these LWP products. Overall, sensitivity of $S_{X\_CDNC}$ to different thresholds using MODIS LWP is more significant than that based




on AMSR-E LWP; this pattern is opposite for $S_{X\_AI}$. It is interesting to note that while the difference in $S_{POP}$ between MODIS and AMSR-E LWP is small with the -15dBZ threshold (Fig. 5; Fig. 9a and Fig. 9c), the difference is relatively larger for the 0 dBZ threshold (Fig. 9a and Fig. 9c), especially at larger LWP bins.

**3.5 $S_{X\_Y}$ from different precipitation data sets**

The diverse rain data sets allow us to explore differences in precipitation susceptibility estimates from different rain products. In Fig. 10, we illustrate $S_{POP}$ and $S_I$ for different rain data sets, namely, 2B-GEOPROF, 2C-PRECIP-COLUMN and 2C-RAIN-PROFILE (Marchand et al., 2008; Haynes et al., 2009; Lebsock and L'Ecuyer, 2011) products. Here we use LWP derived from MODIS and use "rain certain" flag for rain definition reported in the latter two rain products. Since the precipitation flags used in these two rain products are exactly identical (Lebsock and L'Ecuyer, 2011), only $S_{POP}$ based on

2C-PRECIP-COLUMN is plotted in Fig. 10. For 2B-GEOPROF, the threshold of 0dBZ radar reflectivity is used to define a rain event and rain rate is estimated by using $Z=25R^{1.3}$ suggested by Comstock et al. (2004). Note that using "rain certain" flag or threshold of 0 dBZ to identify rain event for those rain products would result in a reduction of rain events across all LWP bins, especially at low LWP bins, therefore we expand bounds of low LWP bins to have enough rain samples at low LWP bins.

$S_{POP}$ exhibits a similar dependence on LWP among these three rain products, but $S_{POP}$ based on 2B-GEOPROF is systematically larger than that based on 2C-PRECIP-COLUMN (this is also true for $S_I$). It is unclear what might lead to higher $S_{POP}$ and $S_I$ from 2B-GEOPROF. The vertical structure of clouds may play a role here, as the maximum radar reflectivity is used from 2B-GEOPROF and surface rain rates are used from the other products.

The most significant discrepancy occurs in $S_{I\_CDNC}$ and $S_{I\_AI}$ (see Fig. 10b). Fig. 10b shows that $S_{I\_CDNC}$ and $S_{I\_AI}$ are

both near-zero for LWP <400 g m$^{-2}$, which may be attributed to high thresholds used among the three rain products. This indicates that precipitation intensity with high threshold is insensitive to CDNC and AI at moderate LWP. This result is consistent with Terai et al. (2015) who suggested that heavy drizzle intensity is insensitive to CDNC. As Fig. 10b shows, $S_{I\_CDNC}$ based on 2C-RAIN-PROFILE product (red squares in Fig. 10b) with subcloud evaporation model incorporated is higher than that based on 2C-PRECIP-COLUMN product (blue squares in Fig. 10b) at high LWP (above 300 gm$^{-2}$). Hill et al.

(2015) showed that, when considering rain evaporation, $S_{I\_CDNC}$ based on surface rain rate is larger than that based on cloud base and column max rain rate at LWP > 400 gm$^{-2}$. However, their difference is more obvious than our results, which may result from threshold used (0.01 mm day$^{-1}$ in Hill et al. (2015) versus surface the 0 dBZ in 2C-RAIN-PROFILE and 2C-PRECIP-COLUMN products). It is interesting to note that the sigh of $S_{I\_CDNC}$ at large LWP is different from that of $S_{I\_AI}$ (Fig. 10b), which is not true for AMSR-E LWP (not shown). This warrants further investigation in the future.



### 3.6 $S_{X\_Y}$ under different stability regimes

Here we examine precipitation susceptibility under different atmospheric stability regimes, as aerosol-cloud-precipitation interactions have been shown to be different under different stability regimes (e.g., L'Ecuyer et al., 2009; Zhang et al., 2016; Michibata et al., 2016). Based on MODIS LWP and 2B-GEOPROF product with -15 dBZ threshold, Fig. 11a and Fig. 11b suggest that both $S_{POP}$ and $S_I$ increase with more stable environment. This pattern for $S_{POP}$ is consistent with the findings of L'Ecuyer et al. (2009) who showed that suppression of precipitation was largest at lower LWP in stable environments. Terai et al. (2015) also found maximum $S_{POP\_CDNC}$ occurred in regions where stable regime was predominant. The distribution of the precipitation susceptibility with respect to LTSS and LWP shown in Fig. 12 using 2B-GEOPROF product with the -15 dBZ rain threshold is consistent with Fig. 11a and Fig. 11b: $S_{POP}$ increases with increasing LTSS with the exception of high LWP. Although not shown here, $S_{POP\_AI}$ based on 2C-PRECIP-COLUMN and AMSR-E LWP product provides a similar pattern with the result of L'Ecuyer et al. (2009), who showed the slope between POP and AI is small both at low and high LWP, but this magnitude tends to increase at intermediate LWP and high LTSS.

Rain definition significantly affects spread of $S_{POP}$ and $S_I$ under different stability regimes. As rain threshold increases, the discrepancy in $S_{POP}$ among different LTSS conditions is more significant (Fig. 11c versus Fig. 11a) while discrepancy in $S_I$ becomes smaller. LTSS-dependence of $S_I$ is even reversed at low LWP with the 0 dBZ threshold compared to that using the -15 dBZ threshold (Fig. 11d versus Fig. 11b).

The above-mentioned features of LTSS-dependency are also true in terms of LWP-weighed mean value, as shown in Fig. 13. For all those cases based on different rain products and LWP products, the LWP-weighed mean of $S_{POP}$ is generally larger under stable conditions compared with unstable conditions. Yet, this feature does not hold true for $S_I$ except the case based on the 2B-GEOPROF dataset with the -15 dBZ threshold.

### 4 Discussion

Fig. 14 shows the range of precipitation susceptibility estimated from different LWP and rain products. Here the threshold of 0dBZ of maximum radar reflectivity is used for 2B-GEOPROF product and the "rain certain" flag is used for 2C-PRECIP-COLUMN and 2C-RAIN-PROFILE products. It shows that uncertainties in $S_{POP}$ (Fig. 14a) as a result of using different LWP and/or rain products are smaller than the uncertainties associated with $S_I$ and $S_R$ (Fig. 14b and c). The uncertainties in $S_{POP}$ are mainly attributed to different LWP products as described in Section 3.4 (see red symbols in Fig. 9a and Fig. 9c).

The results presented here show that "the descending branch" (S decreases with increasing LWP) and "the ascending branch" (S increases with increasing LWP) of $S_{I\_CDNC}$ noted in Feingold et al. (2013) is not only subject to LWP or rain products (Fig. 15b), but also to thresholds chosen to define a rain event (Fig. 9b and Fig. 9d). Therefore, LWP and rain



products themselves play a nonnegligible role when accounting for discrepancy in $S_{I\_CDNC}$ in existing studies. In addition, both $S_{POP}$ and $S_I$ are generally sensitive to the rain threshold choice (Fig. 9).

Interestingly, $S_I$ tends to be negative at low LWP both for AMSR-E and MODIS LWP (Fig. 5b). This is closely associated with positive correlation between conditional-mean rainfall intensity and CDNC (AI) at low LWP bins where

CDNC (AI) is high (Fig. 7e-7h). More negative values are captured when we estimate $S_I$ using 2C-PRECIP-COLUMN and 2C-RAIN-PROFILE products (Fig. 10b and Fig. 13). Furthermore, $S_{I\_CDNC}$ based on these rain products is all negative at low and intermediate LWP regardless of the LWP dataset used. Terai et al. (2015) also found negative values of $S_{I\_CDNC}$ at low LWP and high CDNC. In their study, sign and/or magnitude of $S_{I\_CDNC}$ at low LWP are distinct across different regions. In addition, Koren et al. (2014) found a positive relationship between AOD and rain rate over pristine areas with warm and

aerosol-limited clouds, which was attributed to aerosol invigoration effect. As $S_I$ shows large differences under different stability regimes (Fig. 13), it would be highly interesting to analyze regional variation in $S_I$ to further understand negative $S_I$ in the future, especially under unstable regimes.

Furthermore, our results show that drizzle intensity is more susceptible to aerosol perturbations than rain intensity (see Fig. 9b and Fig. 9d), which might help to explain why negative values of $S_{I\_CDNC}$ occur more frequently with increasing

rainfall thresholds. Jung et al. (2016) found more negative values of $S_{I\_CDNC}$ with increasing threshold (see Fig. B2 in Jung et al. (2016)). In addition, rain products used in our study are all derived from CPR onboard CloudSat. With increasing thresholds, rainfall becomes heavy and uncertainty in rain rate retrieval can grow as CPR is insensitive to heavy precipitation (Haynes et al., 2009). So combination of different rain satellite products (e.g., CloudSat and TRMM) would be helpful for better understanding negative $S_I$.

**5 Summary**

In this paper, we estimate precipitation susceptibility on warm clouds over global oceans based on multi-sensor aerosol and cloud products from the A-Train satellites, including MODIS, AMSR-E, CALIOP and CPR observations, covering the period June 2006 to April 2011. In addition to different aerosol, cloud and rain products, we also analyze other factors that have potential influence on susceptibility, such as different definitions of precipitation susceptibility (six different

susceptibilities defined by Eq. (6)), stability regimes, and thresholds for defining a rain event (e.g., -15dBZ and 0dBZ of maximum radar reflectivity for 2B-GEOPROF). The primary goal of the study is to quantify uncertainties in precipitation susceptibility estimates from satellite observations.

In general, $S_{POP}$ is a relatively robust metric throughout different LWP and rain products and its estimate is less sensitive to different datasets used (Fig. 13-14). $S_{POP\_CDNC}$ shows overall a monotonic decreasing trend with respect to LWP.

$S_{POP\_AI}$ increases to a maximum at low LWP and then decreases with higher LWP. In contrast, $S_I$ differs considerably among





different LWP and rain products (Fig. 13-14). Interestingly, $S_{I\_CDNC}$ and $S_{I\_AI}$ differ between those LWP products with opposite pattern: $S_{I\_CDNC}$ based on MODIS LWP is higher than that using AMSR-E LWP and the reverse is true for $S_{I\_AI}$ (Fig. 13).

Precipitation susceptibility for drizzle (with -15 dBZ rainfall threshold) is significant different from that for rain (with 0

dBZ rainfall threshold) (Fig. 9 and Fig. 13). Our results suggest that onset of drizzle is not as readily suppressed in warm clouds as rainfall (i.e., $S_{POP}$ is significant larger for rain than for drizzle, especially at moderate LWP, Fig. 9). On the other hand, precipitation intensity susceptibility is generally smaller for rain than for drizzle. In addition, the extent of these differences between drizzle and rain depends on the LWP products used.

$S_{X\_AI}$ based on aerosol products at different spatial resolutions (i.e., 10 km versus 1 degree) is consistent with each other.

Chen et al. (2014) also found that aerosol indirect forcing derived from satellite observations was similar from AI observations at different resolutions (i.e., 20 km versus 1 degree). This suggests that aerosol layers over oceans are relatively homogeneous, implying that aerosol properties at coarse resolution may be suitable for studying aerosol-cloud interactions over oceans.

$S_{POP}$ strongly depends on LTSS, with larger value under more stable environment. This dependence is more significant

for rain than for drizzle (Fig. 11 and Fig. 13). These features, however, are less robust for $S_I$ throughout different LWP and rain products as $S_I$ estimates show large uncertainties from different datasets (Fig. 13). Only in the case of $S_I$ estimated from 2B-GEOPROF product for drizzle (with -15 dBZ threshold), does the LTSS-dependence of $S_I$ hold for both MODIS and AMSR-E LWP.

The results presented here show that the discrepancy in magnitude between $S_{X\_AI}$ and $S_{X\_CDNC}$ can be mainly attributed

to the dependency of CDNC on AI. On the global scale, our results show that $S_{X\_AI}$ is about one-third of $S_{X\_CDNC}$. This relationship is more applicable to $S_{POP}$, and is less applicable to $S_I$. In addition, $S_R \approx S_I + S_{POP}$ is generally true for different LWP products and over different LTSS conditions.

It is interesting to note that $S_I$ is negative at low LWP. More negative values are found when $S_I$ is calculated based on 2C-PRECIP-COLUMN and 2C-RAIN-PROFILE products. Our results show that these negative values of $S_I$ may be related

to stability regimes and further show that $S_I$ based on rain samples (with 0 dBZ threshold) tends to be negative. Further studies (regional variation in $S_I$, combination of different rain satellite products, etc.) are needed to understand this issue.

**Acknowledgement**

M. Wang was supported by the National Natural Science Foundation of China (41575073 and 41621005) and by the Jiangsu Province Specially-appointed professorship grant, the One Thousand Young Talents Program. MYD08_D3 and MYD04_L2

products are available through LAADS, the Level 1 and Atmosphere Archive and Distribution System (https://ladsweb.modaps.eosdis.nasa.gov). MYD06_L2 and 2B-GEOPROF data, both collocated to CALIOP subtrack, are




obtained from ICARE Data and Services Center (http://www.icare.univ-lille1.fr/projects/calxtract/products). 2C-PRECIP-COLUMN and 2C-RAIN-PROFILE data sets are available from CloudSat Data Processing Center (http://cloudsat.atmos.colostate.edu/data). CAL_LID_L2_05kmALay data is gained from ASDC, Atmospheric Science Data Center (https://eosweb.larc.nasa.gov). AMSR-E/Aqua L2B Global Swath Ocean product can be obtained from NASA

Distributed Active Archive Center (DAAC) at NSIDC (http://nsidc.org/daac).

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





**Table 1. The summary of previous satellite studies for estimating precipitation susceptibility.**

| Studies | Rain variables | Aerosol Proxies | Thresholds | Behavior | Satellite datasets |
|---|---|---|---|---|---|
| Sorooshian et al., 2009 | I | AI | surface 1mm h$^{-1}$ | $S_I$: ↗ ↘ | 2C-PRECIP-COLUMN AMSR-E L2B-Ocean MYD08-D3 |
| Wang et al., 2012 | POP | AI | rain certain[a] | $S_{POP}$<0.2 | 2C-PRECIP-COLUMN AMSR-E L2B-Ocean MYD08-D3 |
| Terai et al., 2015 | R/POP/I | CDNC | -15dBZ of $Z_{max}$[b] | $S_R$: ↘ $S_{POP}$: ↘ $S_I$: ↗ | 2B-GEOPROF MYD06_L2 |
| Michibata et al., 2016 | I | CDNC | -15dBZ of $Z_{max}$[b] | $S_I$: ↗ ↘ | 2B-GEOPROF MYD06_L2 |

[a]Rain certain is a flag of 2C-PRECIP-COLUMN product, which is equivalent to greater than attenuation-corrected reflectivity threshold of 0dBZ.

[b]$Z_{max}$: the maximum column radar reflectivity from the 2B-GEOPROF product. Symbols of ↗ (↘) represent the increasing (decreasing) trend of susceptibility with increasing LWP.



**Table 2. Satellite products employed to estimate aerosol and cloud properties in this study**

| Parameter | Product | Subset | Horizontal resolution | Sensor | Satellite |
|---|---|---|---|---|---|
| AI | MYD08_D3 | Aerosol_Optical_Depth_Land_Ocean_Mean | 1 ° | MODIS | AQUA |
| | | Aerosol_AE1_Ocean_JHisto_vs_Opt_Depth | | | |
| | MYD04_L2 | Optical_Depth_Land_And_Ocean | 10km | | |
| | | Angstrom_Exponent_1_Ocean | | | |
| | CAL_LID_L2_05kmALay | Column_Optical_Depth_Aerosol_532 | 5km | CALIOP | CALIPSO |
| | | Column_Optical_Depth_Aerosols_1064 | | | |
| CDNC/LWP | MYD06_L2[a] | Cloud_Effective_Radius | 5km | MODIS | AQUA |
| | | Cloud_Optical_Thickness | | | |
| LWP | AE_Ocean_L2B | High_res_cloud | 12km | AMSR-E | AQUA |
| POP/R | 2B-GEOPROF[a] | CPR_Cloud_mask | 5km | CPR | CloudSat |
| | | Radar_Reflectivity | | | |
| | 2C-PRECIP-COLUMN | Precip_rate | 1.4km×1.7km | | |
| | | Precip_flag | | | |
| | 2C-RAIN-PROFILE | Rain_rate | | | |
| | | Precip_flag | | | |

[a]The original horizontal resolution of MYD06_L2 and 2B-GEOPROF products is 1km and 1.4km×1.7km, respectively. Since these products both are obtained from caltrack product collocated to CALIOP subtrack, the resolution is resampled to 5km. Detailed information is provided by the website (http://www.icare.univ-lille1.fr/projects/calxtract/products).





**Table 3. The LWP weighted-mean values of precipitation susceptibility Sx_y and dlnCDNC/dlnAI over global oceans under different stability regimes. The statistics is based on 2B-GEOPROF/CPR product using cloud base Z-R relationship and -15dBZ threshold.**

| | | $S_{R\_AI}$ | $S_{I\_AI}$ | $S_{POP\_AI}$ | $S_{R\_CDNC}$ | $S_{I\_CDNC}$ | $S_{POP\_CDNC}$ | dlnCDNC/dlnAI |
|---|---|---|---|---|---|---|---|---|
| MODIS LWP | global | 0.05 | -0.01 | 0.08 | 0.74 | 0.47 | 0.44 | 0.28 |
| | unstable | -0.04 | -0.07 | 0.04 | 0.52 | 0.34 | 0.26 | 0.22 |
| | stable | 0.23 | 0.14 | 0.15 | 0.83 | 0.52 | 0.67 | 0.30 |
| | midstable | 0.02 | -0.03 | 0.07 | 0.66 | 0.43 | 0.35 | 0.29 |
| AMSR-E LWP | global | 0.17 | 0.08 | 0.11 | 0.47 | 0.19 | 0.37 | 0.32 |
| | unstable | 0.14 | 0.08 | 0.08 | 0.21 | 0.06 | 0.18 | 0.25 |
| | stable | 0.27 | 0.17 | 0.17 | 0.74 | 0.33 | 0.62 | 0.33 |
| | midstable | 0.13 | 0.05 | 0.10 | 0.40 | 0.15 | 0.29 | 0.34 |





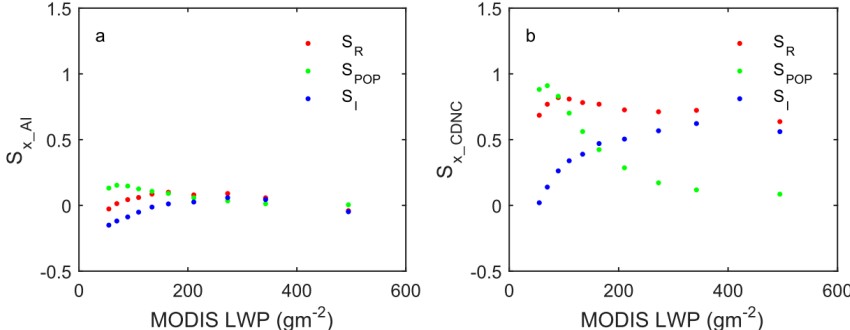

**Figure 1.** $S_{POP}$, $S_I$ and $S_R$ as a function of MODIS LWP with (a) AI and (b) CDNC. AI is derived from MYD04/MODIS and CDNC is estimated from MYD06/MODIS. Intensity and probability of precipitation are based on 2B-GEOPROF product with -15dBZ threshold.




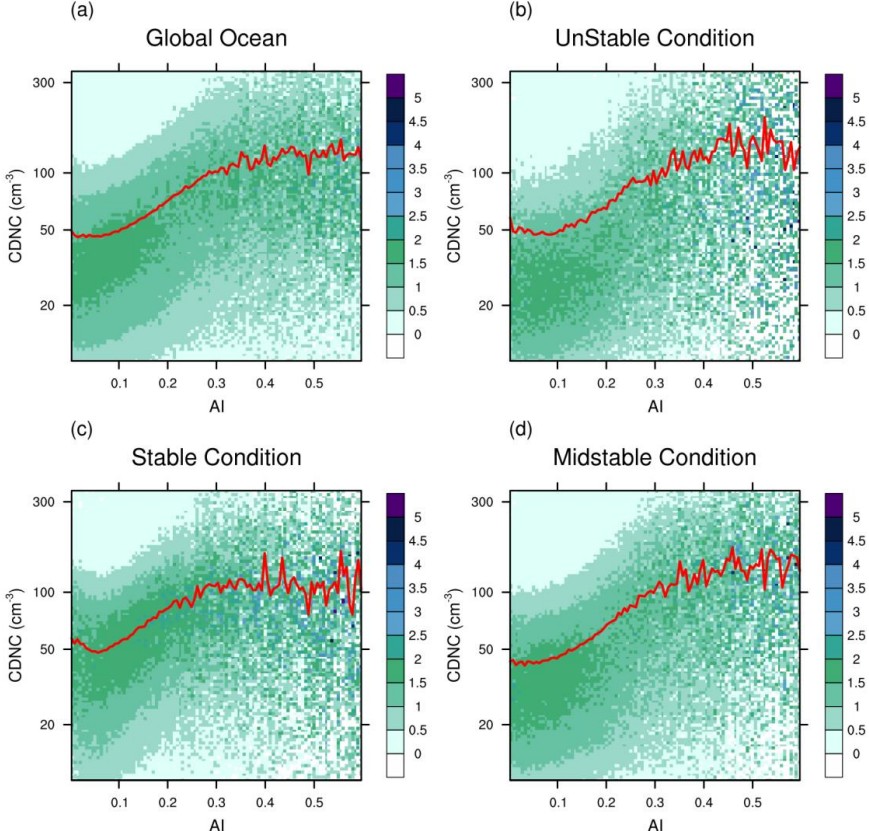

**Figure 2. The probability of CDNC under given AI over (a) global ocean, (b) unstable, (c) stable and (d) mid-stable conditions. In each figure, the red line represents change in average CDNC with AI. AI and CDNC are estimated from MYD04 and MYD06, respectively.**





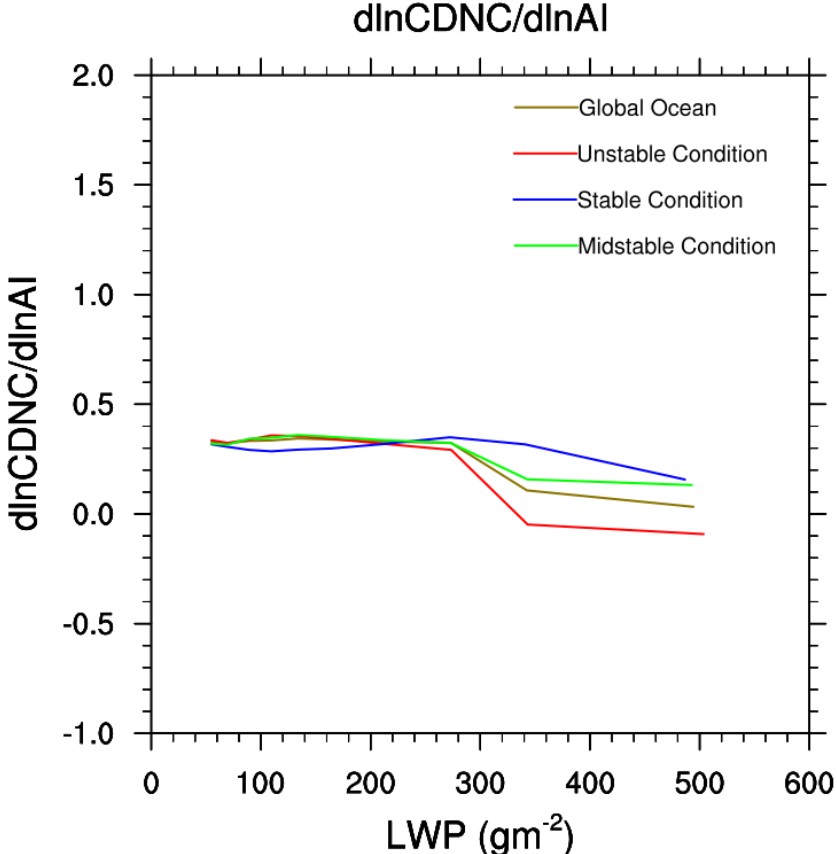

**Figure 3. dlnCDNC/dlnAI obtained by linear regression of lnCDNC and lnAI under MODIS LWP bins. Brown line denote global ocean. Red, blue and green stand for unstable, stable and mid-stable condition, respectively. AI and CDNC are estimated from MYD04 and MYD06.**





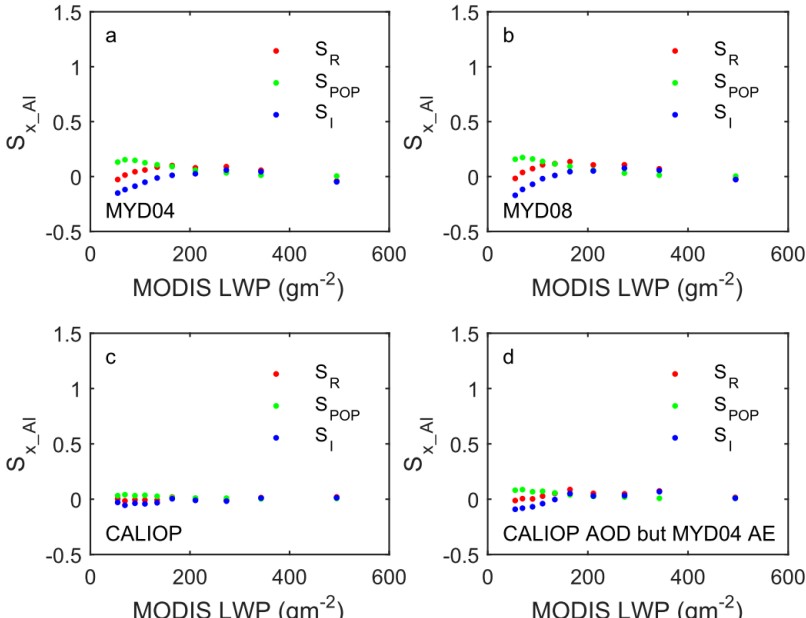

**Figure 4. Susceptibilities ($S_{X\_AI}$) as a function of MODIS LWP. Rain product used is the same as Figure 1. AI is derived from (a) MYD04/MODIS, (b) MYD08/MODIS and (c) CAL_LID_L2_05kmALAy/CALIOP product. Panel d is the same as panel c but using MYD04 AE.**





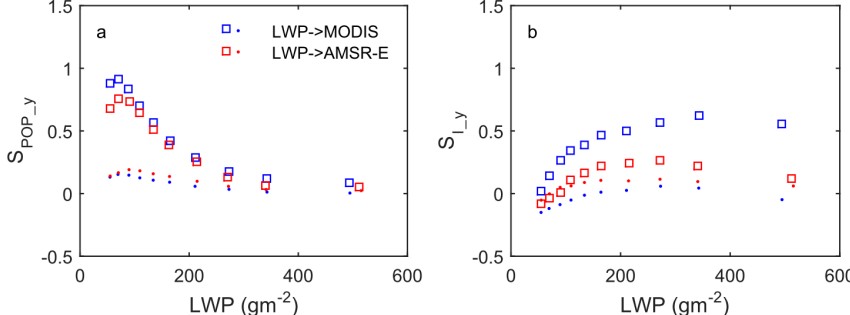

**Figure 5. (a) $S_{POP\_Y}$ and (b) $S_{L\_Y}$ as a function of LWP. The subscript y denotes different aerosol proxies corresponding to AI (point) and CDNC (square). Blue (red) represent LWP derived from MODIS (AMSR-E). Rain product used is the same as Figure 1. AI and CDNC are estimated from MYD04 and MYD06, respectively.**





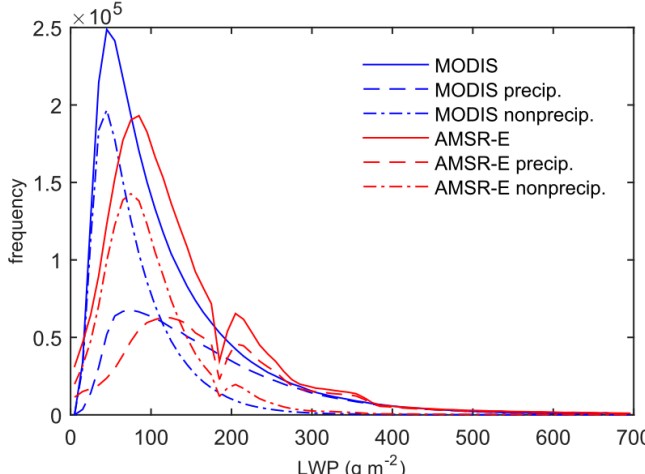

**Figure 6. Distribution of frequency of LWP derived from MODIS and AMSR-E under different scenarios, namely, all samples, nonprecipitation and only precipitation samples.**





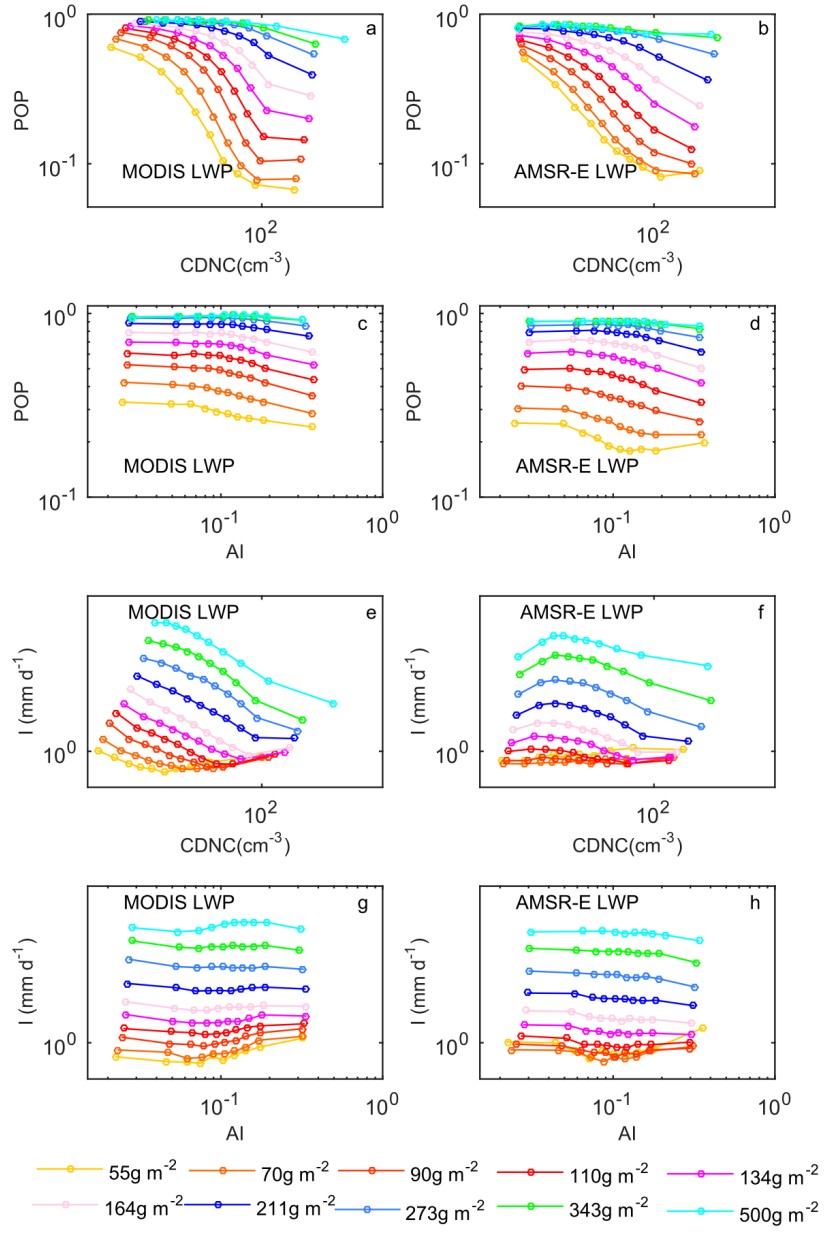

**Figure 7. POP and I as a function of CDNC (AI) for each LWP bin obtained from (left) MODIS and (right) AMSR-E. The data used here is the same as Figure 5.**



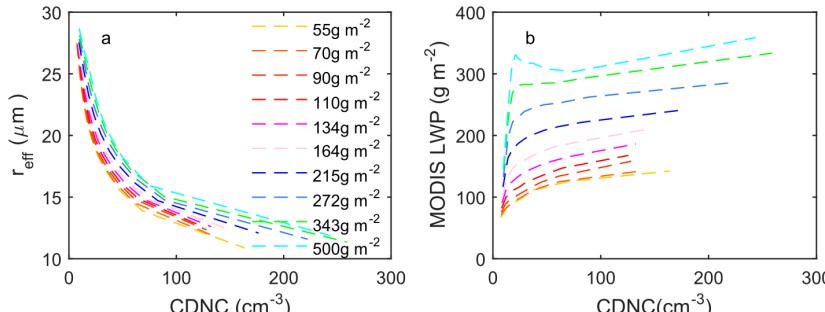

**Figure 8. (a) r$_{eff}$ and (b) MODIS LWP as a function of CDNC for each AMSR-E LWP bin. Only rainy samples defined by -15dBZ threshold are used in here. Different color lines represent different AMSR-E bins corresponding to Fig. 7f.**



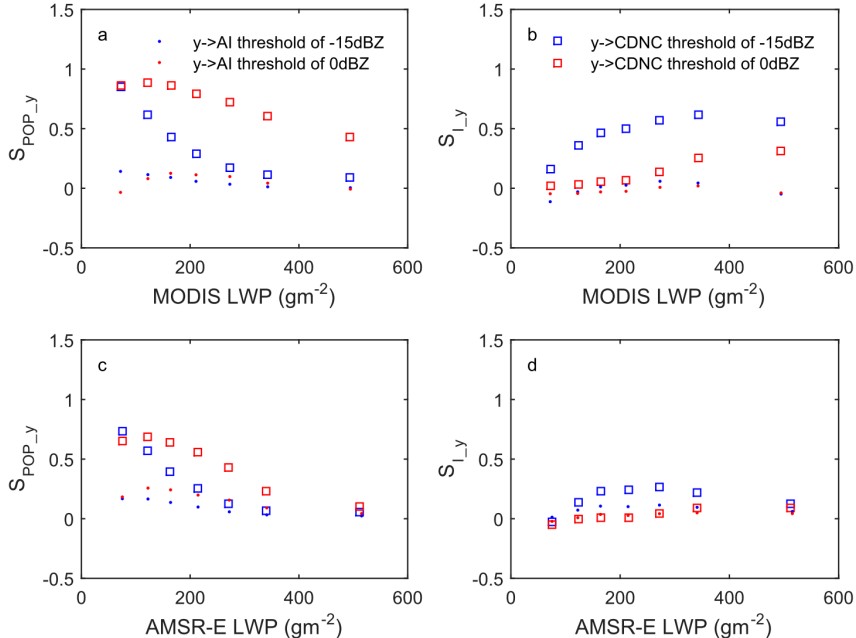

**Figure 9. (a, c) S$_{POP\_Y}$ and (b, d) S$_{I\_Y}$ as a function of LWP. The subscript y denotes different aerosol proxies corresponding to AI (point) and CDNC (square). 2B-GEOPROF product is used here. Blue and red symbols represent -15dBZ threshold and 0dBZ threshold, respectively.**




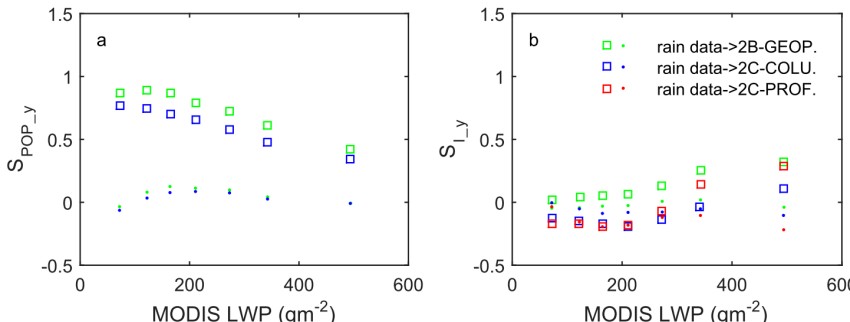

**Figure 10. (a) $S_{POP\_Y}$ and (b) $S_{I\_Y}$ as a function of MODIS LWP. The subscript y denotes different aerosol proxies corresponding to AI (point) and CDNC (square). Different color symbols stand for different rain products: 2B-GEOPROF(2B-GEOP, green), 2C-PRECIP-COLUMN (2C-COLU, blue) and 2C-RAIN-PROFILE (2C-PROF, red). See text for further details.**





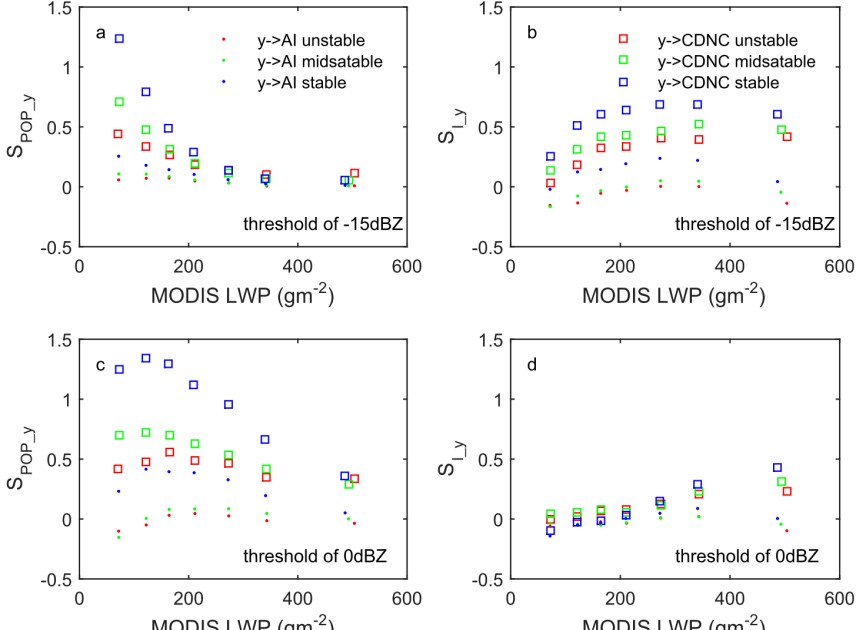

**Figure 11. (a, c)** $S_{POP\_Y}$ **and (b, d)** $S_{L\_Y}$ **as a function of MODIS LWP. The subscript y denotes different aerosol proxies corresponding to AI (point) and CDNC (square). Blue, red and green symbols stand for stable, unstable and mid-stable regimes, respectively. Rain data comes from 2B-GEOPROF. The top panels are for results based on the rain threshold of -15 dBZ and the bottom panels are based on the rain threshold of 0 dBZ.**





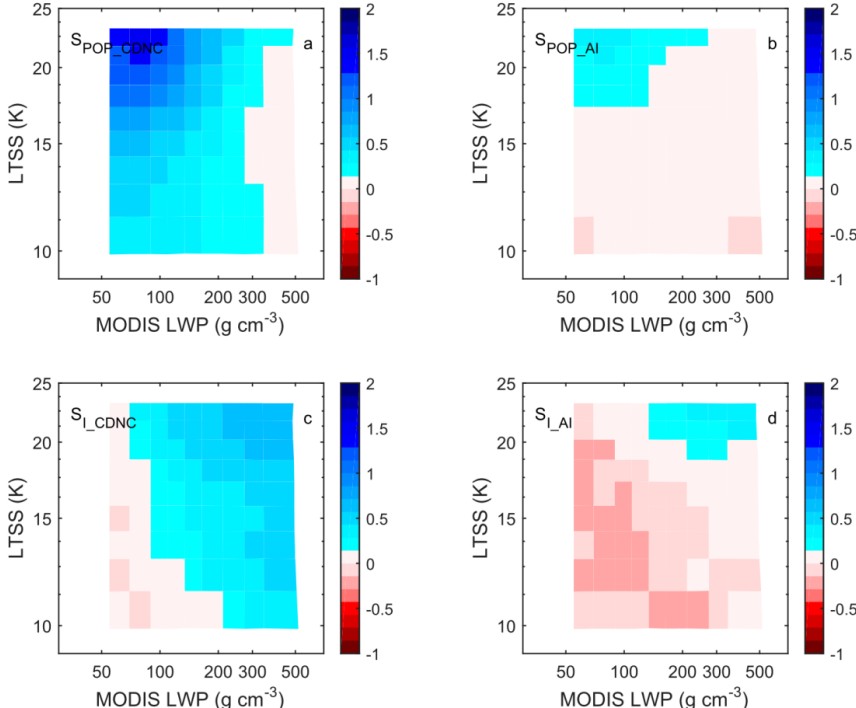

**Figure 12. Distribution of (a-b) $S_{POP\_y}$ and (c-d) $S_{I\_y}$ as a function of MODIS LWP and LTSS. Rain data is from 2B-GEOPROF with threshold of -15dBZ.**





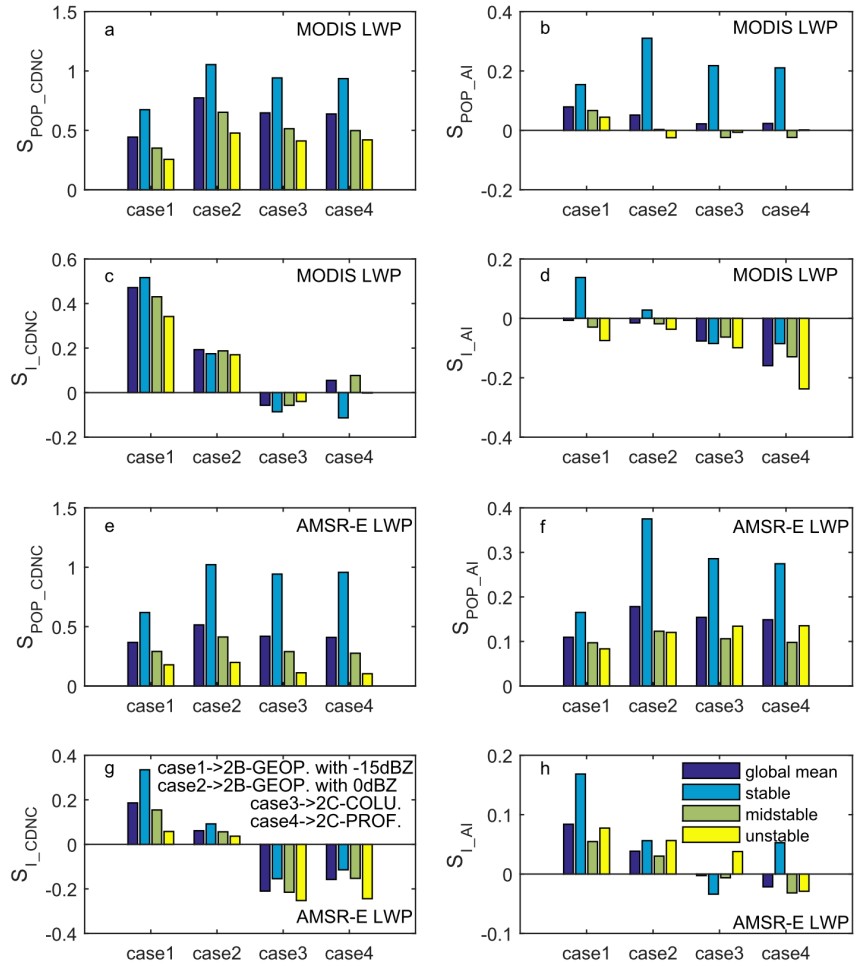

**Figure 13. The LWP-weighed mean values of (a-b, e-f) $S_{POP}$ and (c-d, g-h) $S_I$ under different stability regimes for four cases. The case1 and case2 are both based on 2B-GEOPROF product, but use threshold of -15 dBZ and 0 dBZ, respectively. The case3 and case4 use 2C-PRECIP-COLUMN and 2C-RAIN-PROFILE products, respectively. The top two panels use MODIS LWP and the bottom two panels use AMSR-E LWP.**





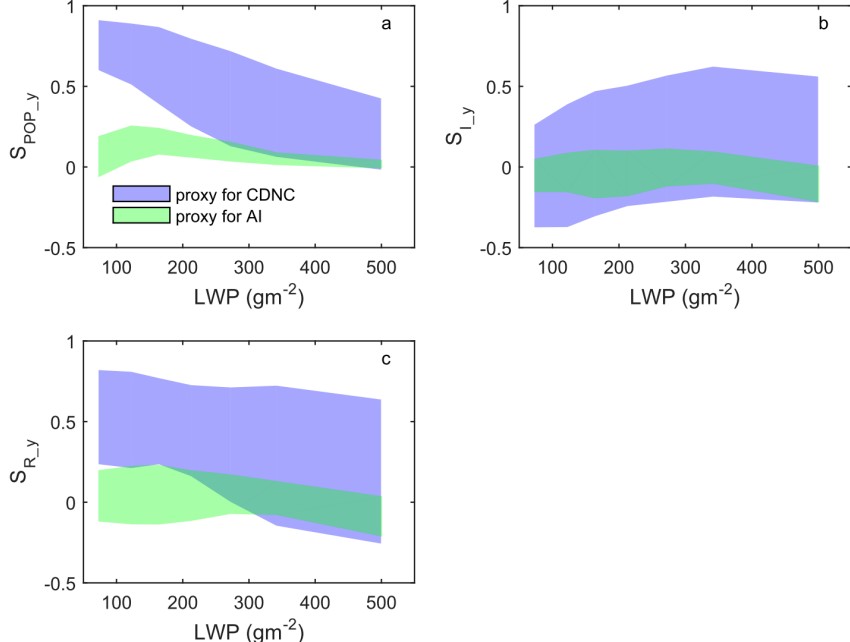

**Figure 14.** (a) $S_{POP\_Y}$, (b) $S_{I\_Y}$ and (c) $S_{R\_Y}$ as a function of LWP. The subscript y denotes different aerosol proxies corresponding to AI (light green) and CDNC (light blue). Shade areas show the range of precipitation susceptibility from different rain products (same as the Fig. 10) and different LWP products (MODIS and AMSR-E).