# Peer review of "Estimating precipitation susceptibility in warm marine clouds using multi-sensor aerosol and cloud products from A-Train satellites"

_Atmospheric Chemistry and Physics, 2017_

## Referee Comment (RC1) · J. Quaas (Referee) · 7 Nov 2017

Bai et al. present a comprehensive calculation of statistics of precipitation vs. aerosol index/cloud droplet concentration from A-Train satellite retrievals. The document differences in the linear regression metrics when applying different (microwave vs. visible-near infrared) LWP retrievals; different precipitation retrievals (albeit all from CloudSat), different aerosol metrics (aerosol index from different retrievals vs. cloud droplet number concentration estimates), and different thresholds.

Although this work does not provide breakthrough science itself, documenting these differences in a consistent way is useful to the debate. The study is in general per-

formed diligently and is pertinent to ACP.

I have two main modifications I recommend, and several specific comments.

I have not highlighted semantic or orthographic mistakes since I assume the Copernicus copy editing will take care of this.

Main remarks

1) The authors should expand their discussion of the state of the art, especially they need to discuss the various other aerosol-precipitation interactions beyond the "lifetime effect". It is in particular necessary that the authors discuss the role of aerosol scavenging when interpreting the metrics they investigate.

2) I am a bit astonished on how rather poorly the figures are done. The authors should take care revising these so that the content is more readily understandable.

Specific remarks

P1 l15: Here and at a plenty of instances in the text, the relationship between aerosol and precipitation derived from the observations is overly readily interpreted in a cause-effect manner. If only this science was so easy, then a plenty of issues wouldn't exist. I urge the authors to thoroughly revise their text and imply causality only where they can prove it, or at least where they can corroborate cause-effect relationships. Why not interpret a negative aerosol index – POP relationship, for example, as showing the wet scavenging precipitation effect on aerosol?

P1 l26: I suggest the authors adapt to the IPCC AR5 language and define the radiative forcing due to aerosol-cloud interactions ("cloud albedo effect") and cloud adjustments (all subsequent modifications). It is necessary that the authors put the "cloud lifetime effect" hypothesis into context of the manifold other hypotheses.

P2 l4: Also, the relationship would need to be linear (or more generally, of known, universal, monotonic functional form).

P4 l17: But L3 is at 1°, so far from the stated 5 km resolution

p4 l25: again, why the two, and not only the L2 data?

P5 l1: The authors should report exactly how the colocation is done.

P5 l5: A discussion of Christensen et al. doi 10.5194/acp-2017-450 would be useful here

p5 l7: Wood and Hartmann is a good paper, but is it a pertinent reference here?

P5 l12: What is a "pixel" here? A 1-km MODIS cloud retrieval, or rather an aggregated CALTRACK 5 km grid box?

P5 l15: Is this statement tested/implemented? Or is it just taken for granted from the Kubar study?

P6 l29: It is nonsense that LTS is able to clearly distinguish cloud regimes (e.g. Nam and Quaas doi 10.1002/grl.50945). Klein and Hartmann only show that the seasonal cycles of cloud fraction and LTS correlate.

P6 l30: The term "unstable" is a misnomer. "unstable" would mean, a negative LTSS.

P7 l10: Why this choice and not deciles?

P9 l5: When using AMSR-E LWP, are the pixels selected overcast at AMSR-E footprint? Or is the AMSR-E interpolated to the CALTRACK grid cells?

P9 l17: Is this maybe due to the fact that AMSR-E LWP in fact is cloud fraction times in-cloud LWP, in combination with the fact that CDNC is positively correlated to cloud fraction (Fig. 2)?

Fig. 1: Since there are only ten bins in LWP, I suggest to label each bin center on the x-axis. Possibly the axis could be chosen irregular then. It would be good to indicate the total amount of data points in the caption. In (a) a zeroline would be helpful.

Fig. 4: The authors need to choose a different y-axis that spans only the range of data.

As it is now, no details can be distinguished. Again, a zeroline is necessary.

Fig. 5: zeroline would be good

Fig. 7: a, b, e, f: more x-axis tick marks necessary; e-f: more y-axis tick marks necessary

Fig. 10 b: zeroline necessary

Fig. 12: the color code is poorly selected. The colors should be centered around zero (light pink shouldn't indicate positive). Are the LTSS bins chosen so that each contains on average the same amount of pixels (that is the way it should be, else a PDF of LTSS would need to be shown).

Fig. 14: is it not possible to differentiate likelihoods, e.g. by putting equal weight on each curve entering the shaded area and then varying the color intensity/darkness?

---

## Referee Comment (RC2) · Anonymous Referee #2 · 11 Nov 2017

In various studies, the precipitation susceptibility metric has been used to quantify the effect of aerosols on the precipitation in both models and observations and to indicate the strength of the cloud lifetime effect. The present article examines how observationally-based estimates of the precipitation susceptibility metric vary depending on the various dataset and analysis choices. Previous attempts to provide an observational constraint on the precipitation susceptibility metric have led to different strengths in susceptibility and also in different behaviors of the susceptibility. The study contributes to the existing literature by attempting to reconcile those differences by examining a wide range of data and analysis choices in the same framework, which might help answer why different studies have arrived at different susceptibility estimates. The

authors examine the sensitivity of the susceptibility metric to the choice of aerosol proxy, precipitation characteristic (intensity, probability of precipitation (POP), or mean precipitation), stability regime, liquid water path retrieval, precipitation retrieval, and precipitation threshold. After examining the whole range of sensitivities, the authors conclude that SPOP has the least amount of spread that arises from the choice of liquid water path and precipitation data product. The authors also find strong sensitivities in the choice of stability regime and in whether aerosol index (AI) or the cloud droplet number concentration (CDNC) is used as the aerosol proxy.

The study is a substantial contribution to the existing literature by providing a comprehensive examination of the possible source of discrepancies that can arise when trying to estimate the precipitation susceptibility based on satellite retrievals. The manuscript methodically goes through the different choices that can be made, and assesses their impact on the value and behavior of the metric. There are a couple issues with the paper that I would like to see the authors address. First, the authors mention in the introduction of how estimates from Wang et al. (2012), Terai et al. (2015), and Michibata et al. (2016) differ in the magnitude of the SPOP metric. Although it appears that the use of AI or CDNC is the largest source of the discrepancy, I expected to see the authors discuss more thoroughly how the effect of the choice of aerosol proxy compares with effect of the choice of precipitation dataset and threshold. I had also expected a similar discussion that folds in the results from Sorroshiaan et al. (2009) on both the magnitude of the susceptibility, as well as the behavior of the susceptibility. Second, statistical confidence limits to the susceptibilities should be provided to determine how the statistical uncertainties compare with the other dataset/methodology uncertainties that are examined in the study. The confidence intervals would help inform whether the choice of datasets significantly change the susceptibility estimates or not. Overall, the manuscript has a clear scientific question, uses analyses that address the question, and is well organized. I do not consider the main issues that I have to be major. Therefore, I recommend that the manuscript be published after the following comments and issues have been addressed.

[Figure]

Main comments and issues

1) The uncertainties in the susceptibility estimates should be reported in all the figures and graphs. The 95% confidence intervals can be calculated from the standard deviation of the regression or using bootstrapping techniques. The statistical uncertainties will help the author substantiate some of the statements within the manuscript that say whether or not various choices significantly change the susceptibilities.

2) Given that the main purpose of the study is to examine how the various choices have led to differences in the susceptibility that are reported in the literature, the authors should provide more discussion on how this study helps to reconcile existing differences. In particular, the authors should do their best to identify likely reasons why the estimates in the previous studies have differed (if they do). For example, there are differences in the magnitude of the susceptibility (e.g., Wang et al., 2012 versus Terai et al., 2015). There are also differences in the behavior of susceptibility (monotonic decrease versus increase and then decrease).

3) The authors seem to argue for the use of SPOP as a metric to quantify aerosol-cloud-precipitation interactions due to SPOP having a smaller range of possible values, based on different LWP and precipitation rate retrievals (Fig. 14). There is less discussion on the advantages and disadvantages of using CDNC or AI as a metric and also a lack of discussion on how the threshold (rain vs. drizzle) can significantly change SPOP values. Given that the authors have examined a wide range of potential sources that lead to differences in susceptibility estimates, it would be informative for the readers to have the authors synthesize their findings and discuss what should be considered in future attempts to try to observationally constrain precipitation susceptibility or attempts to compare susceptibilities from models and from observations.

Minor comments

P2 L3: "Susceptibility is an inherent property of the aerosol-cloud system." – This is an interesting statement, but it is also vague. Does the statement mean that susceptibility

doesn't change with cloud condition? Or aerosol condition? Should they be robust to differences in measurement platform, etc.?

P5 L32: "... selected in close proximity of clouds pixels." What exact criteria is used to determine how close aerosol retrievals must be to be used in the study?

P6 L23: replace "significant" with "significantly"

P6 L25: What is the spatial resolution of the precipitation data? Is it at the footprint level? In general, how are pairs of LWP, precipitation rate, and aerosol proxy combined? Are they all combined at the footprint of the precipitation rate? Is the coarsest footprint used for the comparison?

P7 L15: Provide some indication of statistical uncertainty in the estimates in Fig. 1. See main comment 1. Also, it would be informative to indicate the 0 value with a dotted or gray line, because values below that line will indicate that increases in aerosols/cloud droplets lead to more precipitation.

P7 L22-24: The turning point is very slight. The confidence intervals will be helpful in determining how significant the peak is.

P7 L29: To show that the fluctuations in the mean are small compared to noise, the interquartile range (between 25th percentile and 75th percentile) can be shown.

P7 L29: Also, because the AI vs. CDNC relationship takes the form d ln(CDNC)/d ln(AI) and because it looks like the AI has a lognormal distribution, it might be better to plot the x-axis in log-scale.

P8 L4-5: The differences in dlnCDNC/dlnAI between the different stability regimes is interesting, in particular, the lack of sensitivity (or negative sensitivity at high LWPs). Are these differences significant? Do the authors have an explanation as to why the stability affects the sensitivity?

P8 L16: The subtle differences in Fig 4 are hard to see because of the large y-axis

range. I can understand the choice to try to keep the same axis range across different figures, but in this case, I would suggest narrowing the range to allow the reader to discern any differences.

P8 L26-28: Is there a reason why we would rely more heavily on and prefer MODIS AI rather than CALIPSO AI?

P9 L2-7: This is one case where confidence intervals can show that the SPOP estimates are not significantly affected by the choice of LWP retrievals, whereas the SI and SR estimate are significantly affected.

P9 L26: Data is plural, so it should be "... when data are binned..."

P10 L11: Although the axis labels show this, the figure caption to Figure 9 should indicate the difference between the top row and the bottom row.

P10 L26-28: What is the impact on SR if SPOP increases and SI decreases with increases in the threshold?

P10 L31: "more significant" should be replaced with "larger", because significant has a particular meaning in the literature (statistical significance), and to state more significant would require examining the confidence intervals.

P11 L28: "sigh" should be replaced by "sign"

P14 L5: Insert "by increases in AI or CDNC" between "readily suppressed" and "in warm clouds"

P14 L5-6: Taken at face value, this statement is counterintuitive, isn't it? Wouldn't we expect rainfall, which is more dependent on accretion than on autoconversion, to have a weaker dependence to CDNC?

P14 L14: Replace "value" with "values"

P14 L14-18: Are these results consistent with existing conceptual frameworks (such as

those based on LES) on how stability affects aerosol-cloud-precipitation interactions? Are there LES studies that have addressed how stability might affect susceptibility?

---

## Author Comment (AC1) · 9 Dec 2017

**We are grateful to the evaluations from the reviewers, which have allowed us to clarify and improve the manuscript. Below we addressed the reviewer comments, with the reviewer comments in italic and our response in bold.**

*J.Quaas (Referee #1)*

*Bai et al. present a comprehensive calculation of statistics of precipitation vs. aerosol index/cloud droplet concentration from A-Train satellite retrievals. The document differences in the linear regression metrics when applying different (microwave vs. visible-near infrared) LWP retrievals; different precipitation retrievals (albeit all from CloudSat), different aerosol metrics (aerosol index from different retrievals vs. cloud droplet number concentration estimates), and different thresholds.*

*Although this work does not provide breakthrough science itself, documenting these differences in a consistent way is useful to the debate. The study is in general performed diligently and is pertinent to ACP*

*I have two main modifications I recommend, and several specific comments.*
*I have not highlighted semantic or orthographic mistakes since I assume the Copernicus copy editing will take care of this.*

*Main remarks:*
*1. The authors should expand their discussion of the state of the art, especially they need to discuss the various other aerosol-precipitation interactions beyond the "lifetime effect". It is in particular necessary that the authors discuss the role of aerosol scavenging when interpreting the metrics they investigate.*
**Thanks for your comments! We now added more discussion in Section 4 to acknowledge aerosol-precipitaiton interactions beyond the "lifetime effects", and it reads: "It should be noted that precipitation susceptibility in our study is based on Eq. (7) and is derived by linear regression between precipitation fields and CDNC/AI in log-log space. The negative/positive correlation between precipitation frequency/intensity and aerosols may not be readily explained as aerosol effects on precipitation. For example, a negative correlation between precipitation frequency and aerosols may come from the wet scavenging effects of aerosols (more precipitation leads to less aerosols) but not aerosol suppression of precipitation. However, in our study, we not only calculate precipitation susceptibility with respect to AI ($S_{X\_AI}$), but also with respect to CDNC ($S_{X\_CDNC}$) and the later one is expected to be less affected by the wet scavenging effects. The broad consistency between these two estimates shown in our results (Fig. 13), especially for the estimate of $S_{POP}$, lends the support to the limited**

influence of wet scavenging in our estimate. Further support for this comes from the fact that precipitation susceptibility estimates based on the 1 degree L3 MODIS aerosol products are similar to those based on the 10 km L2 MODIS aerosol products (Fig. 4), as we would expect the wet scavenging effects are more important at smaller scales if the wet scavenging effects are a dominating factor. Nevertheless, the effects of wet scavenging can still be important in satellite studies of aerosol-cloud-precipitation interactions, and should be better quantified in future, perhaps in combination with model simulations.".

*2. I am a bit astonished on how rather poorly the figures are done. The authors should take care revising these so that the content is more readily understandable.*
**Thanks a lot for your suggestions! We have revised most figures in the paper according to the comments provided by two referees. This includes adding 95% confidence intervals and zero lines to most figures, and narrowing the range of y-axis in most figures.**

*Specific remarks*
*1. P1 l15: Here and at a plenty of instances in the text, the relationship between aerosol and precipitation derived from the observations is overly readily interpreted in a cause-effect manner. If only this science was so easy, then a plenty of issues wouldn't exist. I urge the authors to thoroughly revise their text and imply causality only where they can prove it, or at least where they can corroborate cause-effect relationships. Why not interpret a negative aerosol index – POP relationship, for example, as showing the wet scavenging precipitation effect on aerosol?*
**The sentence is now reformulated to "We find that $S_{POP}$ strongly depends on atmospheric stability, with larger values under more stable environments". We have added more discussion in Section 4 for addressing this issue. See our reply to your main remaks#1.**

*2. P1 l26: I suggest the authors adapt to the IPCC AR5 language and define the radiative forcing due to aerosol-cloud interactions ("cloud albedo effect") and cloud adjustments (all subsequent modifications). It is necessary that the authors put the "cloud lifetime effect" hypothesis into context of the manifold other hypotheses.*
**Thanks for this suggestion! We now adapted the IPCC AR5 language in the manuscript. The text in the first paragraph of the introduction now reads "Aerosol-cloud interactions play an important role in the climate system and affect the global energy budget and hydrological cycle. The effective radiative forcing from aerosol-cloud interactions (ERFaci), which includes the instantaneous effect on cloud albedo from changes in cloud condensation nuclei (CCN) or ice nuclei and all subsequent changes to cloud lifetime and thermodynamics, remains one of the largest**

uncertainties in our estimates of anthropogenic radiative forcing (Boucher et al., 2013)." We also removed all other references to "cloud lifetime effect" in the manuscript and replaced it by "cloud water response to aerosol perturbations".

*3. P2 l4: Also, the relationship would need to be linear (or more generally, of known, universal, monotonic functional form).*
**Thanks! We agree and now removed this sentence.**

*4. P4 l17: But L3 is at 1◦, so far from the stated 5 km resolution.*
**We are sorry for the confusion here. We now further clarified in the revised manuscript about how the collocation among different datasets is done (See the first paragraph in the section 2.1), including 1 degree MODIS L3 dataset. The reason why we used L3 aerosol product is because we would like to examine how aerosol homogeneity might affect the estimate of precipitation susceptibility. This is now added in the first paragraph of Section 2.2.1 and it reads "This MODIS Level 3 dataset has been used in previous studies to examine aerosol-cloud-precipitation interactions (e.g., L'Ecuyer et al., 2009; Wang et al., 2012) and is compared here with results form the MODIS Level 2 aerosol product to examine how aerosol homogeneity might affect precipitation susceptibility estimates.". The comparison between the MODIS L3 and L2 products is documented in details in Section 3.2.**

*5. P4 l25: again, why the two, and not only the L2 data?*
**Please see our reply to comment#4.**

*6. P5 l1: The authors should report exactly how the colocation is done.*
**Thanks for the suggestion. We now clarified this in Sec 2.1 and the text reads "MODIS cloud product and CPR radar reflectivity observations used in this study are both provided from the Caltrack datasets, which resample observations from many sensors under CALIOP subtrack with the horizontal resolution of 5km (see the website of http://www.icare.univ-lille1.fr/projects/calxtract/products for more information). For other aerosol and cloud products, including MODIS/CALIOP aerosol products and AMSR-E cloud products, they are further collocated into the CALIOP subtracks in the Caltrack dataset. For each CALIOP subtrack, the closest aerosol/ cloud retrieval sample within one-degree grid box (1°×1°) centered at this subtrack is chosen. To reduce the uncertainty in cloud retrievals, only samples where MODIS cloud fraction is equal to 100% are selected. "**

*7. P5 l5: A discussion of Christensen et al. doi 10.5194/acp-2017-450 would be useful here.*

**Thanks for bringing this paper to our attention! This is highly relevant to our study and this paper is now added and discussed in the revised manuscript and it reads "Retrievals of aerosol properties from passive sensors and lidar observation are both affected by clouds near the aerosol, and thereby result in overestimation for aerosol property (Chand et al., 2012; Tackett and Di Girolamo, 2009; Christensen et al., 2017). The extent of this overestimation may be different among different sensors, and depends on how far aerosol pixels chosen are from the corresponding cloud pixels (Christensen et al., 2017).".**

*8. P5 l7: Wood and Hartmann is a good paper, but is it a pertinent reference here?*
**Thanks! We checked and this is now removed.**

*9. P5 l12: What is a "pixel" here? A 1-km MODIS cloud retrieval, or rather an aggregated CALTRACK 5 km grid box?*
**Pixel here is an aggregated CALTRACK 5 km grid box as MODIS cloud product is provided from the Caltrack datasets. Sorry for the confusion. The sentence is reformulated to "To reduce the uncertainty when deriving CDNC, cloud pixels (identified by Caltrack-MODIS cloud product with the horizontal resolution of 5 km) where cloud optical depth is less than 3 and cloud fraction is less than 100% are excluded (Cho et al., 2015; Zhang and Platnick, 2011)."**

*10. P5 l15: Is this statement tested/implemented? Or is it just taken for granted from the Kubar study?*
**Thanks! We now examined the percentage of single layer clouds in our study and it is 94%, consistent with Kubar et al. (2009), and we now updated the text and it reads: "Additionally, we limit our analysis to warm clouds by screening cloud pixels with cloud top temperature warmer than 273K. Under these screening criteria, our results show that 94% warm clouds are single layered (93% in Kubar et al., 2009). Therefore, our analysis mainly focuses on single-layer clouds."**

*11. P6 l29: It is nonsense that LTS is able to clearly distinguish cloud regimes (e.g. Nam and Quaas doi 10.1002/grl.50945). Klein and Hartmann only show that the seasonal cycles of cloud fraction and LTS correlate.*
**We agreed that it is still a challenging task to find a unique metric to clearly distinguish different cloud regimes. In previous studies, several metrics were applied to define different cloud regimes. For instance, by using LTSS and vertical pressure velocity, Zhang et al., (2016) divided descending regimes into stratocumulus, transitional clouds and trade wind cumulus regimes. Webb et al., (2015) developed an index (ALPI) based on LTSS and precipitation to distinguish cloud regimes.**

LTSS may have its limitation for defining different cloud regimes. However, our results show that precipitation susceptibility has clear LTSS-dependence, especially for $S_{POP}$ (Fig 11 and Fig 13). This suggests LTSS provides a feasible way to examine how precipitation susceptibility may depend on cloud regimes. LTSS was also used in many previous studies (e.g.,L'Ecuyer et al., 2009; Terai et al., 2015). Nevertheless, we acknowledged the limitation of LTSS in the revised manuscript in Section 3.6 and it reads: " Our results also suggest that it is important to account for the influence of atmospheric stability owing to the clear dependence of $S_{POP}$ on metrics like LTSS, though it is acknowledged that LTSS alone is an imperfect metric for isolating cloud regimes (e.g., Nam and Quaas, 2013). Different metrics associated with cloud regimes should be examined in future to better understand the effect of cloud regimes on precipitation susceptibility. For instance, LTSS can be combined with vertical pressure velocity to distinguish between different cloud types (Zhang et al., 2016).".

*12. P6 l30: The term "unstable" is a misnomer. "unstable" would mean, a negative LTSS.*
Given that our study focus on ocean warm clouds mostly with positive LTSS values, we followed the same definition of unstable as L'Ecuyer et al., (2009) and Wang et al., (2012), and they both defined unstable environment by LTSS values less than 13.5K.

*13. P7 l10: Why this choice and not deciles?*
In our analysis, we keep the LWP bins the same when we compare different satellite products in individual plots in order to facilitate the comparison. So the number of samples for each LWP varies, from 5% to 14%. However, each LWP bin still includes more than ten thousand samples, large enough for producing robust estimate of precipitation susceptibility.

*14. P9l5: When using AMSR-E LWP, are the pixels selected overcast at AMSR-E footprint? Or is the AMSR-E interpolated to the CALTRACK grid cells?*
The AMSR-E pixels closest to CALTRACK grid cells are selected. We do not require the AMSR-E pixels to be overcast, but clouds from the CALTRACK pixels are overcast with MODIS cloud fraction of 100%. The details of collocation strategy are added to Sec 2.1 in the revised manuscript.

*15. P9 l17: Is this may be due to the fact that AMSR-E LWP in fact is cloud fraction times in-cloud LWP, in combination with the fact that CDNC is positively correlated to cloud fraction (Fig. 2)?*
Thanks! If this is the case, for a constant AMSR-E LWP shown in Fig. 7f, in-cloud LWP would decrease with increasing CDNC as increasing CDNC means increasing cloud fraction. Smaller in-cloud LWP would then imply lower precipitation intensity, opposite to what is shown in Fig. 7f. Our

results shown in Fig. 8 suggests that this might be related to differences in MODIS and AMSR-E LWP at low MODIS CDNC, but what might cause the discrepancies in two LWP product still needs further investigation in the future.

*16. Fig. 1: Since there are only ten bins in LWP, I suggest to label each bin center on the x-axis. Possibly the axis could be chosen irregular then. It would be good to indicate the total amount of data points in the caption. In (a) a zeroline would be helpful.*
**We have added a zeroline in the Fig. 1. The total number of data points now is included in the caption. Since labels of the x-axis would be dense and overlapped at low LWP if we label each mean value of LWP bin, the x-axis now is divided into smaller intervals. In addition, each mean value of LWP bin is shown in Fig.7.**

*17. Fig. 4: The authors need to choose a different y-axis that spans only the range of data. As it is now, no details can be distinguished. Again, a zeroline is necessary*
**The range of y-axis is now narrowed and a zeroline is also added for most figures (Fig.1, Fig.3-Fig.5 and Fig.9-Fig.11)in the revised manuscript.**

*18. Fig. 5: zeroline would be good*
**A zeroline is now added in the Fig.5.**

*19. Fig. 7: a, b, e, f: more x-axis tick marks necessary; e-f: more y-axis tick marks necessary*
**More x-axis and y-axis tick marks are now added accordingly.**

*20. Fig. 10 b: zeroline necessary*
**A zeroline is now added in the Fig.10.**

*21. Fig. 12: the color code is poorly selected. The colors should be centered around zero (light pink shouldn't indicate positive). Are the LTSS bins chosen so that each contains on average the same amount of pixels (that is the way it should be, else a PDF of LTSS would need to be shown).*
**We have changed the color code of Fig. 12 and its colors are now centered around zero. The light pink now indicates negative value. In this figure, each LTSS bin now contains on average the same amount of pixels. We also have added this sentence to the caption of Fig. 12.**

*22. Fig. 14: is it not possible to differentiate likelihoods, e.g. by putting equal weight on each curve entering the shaded area and then varying the color intensity/darkness?*
**Thanks a lot for your suggestion! We would like to take this suggestion, but as we only have eight curves for each metric shown in Fig. 14, we do not**

have large number of curves to show the likelihoods., so we have to keep the figure as it is.

---

## Author Comment (AC2) · 9 Dec 2017

**We are grateful to the evaluations from the reviewers, which have allowed us to clarify and improve the manuscript. Below we addressed the reviewer comments, with the reviewer comments in italic and our response in bold.**

*Anonymous Referee #2*

*In various studies, the precipitation susceptibility metric has been used to quantify the effect of aerosols on the precipitation in both models and observations and to indicate the strength of the cloud lifetime effect. The present article examines how observationally-based estimates of the precipitation susceptibility metric vary depending on the various dataset and analysis choices. Previous attempts to provide an observational constraint on the precipitation susceptibility metric have led to different strengths in susceptibility and also in different behaviors of the susceptibility. The study contributes to the existing literature by attempting to reconcile those differences by examining a wide range of data and analysis choices in the same framework, which might help answer why different studies have arrived at different susceptibility estimates. The authors examine the sensitivity of the susceptibility metric to the choice of aerosol proxy, precipitation characteristic (intensity, probability of precipitation (POP), or mean precipitation), stability regime, liquid water path retrieval, precipitation retrieval, and precipitation threshold. After examining the whole range of sensitivities, the authors conclude that SPOP has the least amount of spread that arises from the choice of liquid water path and precipitation data product. The authors also find strong sensitivities in the choice of stability regime and in whether aerosol index (AI) or the cloud droplet number concentration (CDNC) is used as the aerosol proxy.*

*The study is a substantial contribution to the existing literature by providing a comprehensive examination of the possible source of discrepancies that can arise when trying to estimate the precipitation susceptibility based on satellite retrievals. The manuscript methodically goes through the different choices that can be made, and assesses their impact on the value and behavior of the metric. There are a couple issues with the paper that I would like to see the authors address. First, the authors mention in the introduction of how estimates from Wang et al. (2012), Terai et al. (2015), and Michibata et al. (2016) differ in the magnitude of the SPOP metric. Although it appears that the use of AI or CDNC is the largest source of the discrepancy, I expected to see the authors discuss more thoroughly how the effect of the choice of aerosol proxy compares with effect of the choice of precipitation dataset and threshold. I had also expected a similar discussion that folds in the results from Sorroshiaan et al. (2009) on both the magnitude of the susceptibility, as well as the behavior of the susceptibility. Second, statistical confidence limits to the susceptibilities should be provided to determine how the statistical uncertainties compare with the other dataset/methodology uncertainties that are*

*examined in the study. The confidence intervals would help inform whether the choice of datasets significantly change the susceptibility estimates or not. Overall, the manuscript has a clear scientific question, uses analyses that address the question, and is well organized. I do not consider the main issues that I have to be major. Therefore, I recommend that the manuscript be published after the following comments and issues have been addressed.*

*Main comments and issues:*
*1. The uncertainties in the susceptibility estimates should be reported in all the figures and graphs. The 95% confidence intervals can be calculated from the standard deviation of the regression or using bootstrapping techniques. The statistical uncertainties will help the author substantiate some of the statements within the manuscript that say whether or not various choices significantly change the susceptibilities.*

**The error bars with 95% confidence intervals are now added to all the susceptibilities figures except Fig. 10 and Fig. 11. Given that each panel of Fig. 10 and Fig. 11 includes six susceptibility curves, these figures would be not clear and messy if error bars are added. The error bars can be found for global mean values for these cases in Fig. 13.**

**We thank the reviewer for this excellent suggestion! Adding the statistical uncertainties indeed helps us substantiate some of our statements in the manuscript. For instance, we can state with confidence that $S_{POP}$ estimates are not significantly influenced by LWP products, while $S_I$ estimates are, as shown in Fig. 5 in the revised manuscript. On the other hand, the differences of dlnCDNC/dlnAI between different stability regimes are not significant (Fig. 3). We also find that almost all of mean $S_{I\_CDNC}$ is significantly negative regardless of stability regimes (Fig. 13).**

*2. Given that the main purpose of the study is to examine how the various choices have led to differences in the susceptibility that are reported in the literature, the authors should provide more discussion on how this study helps to reconcile existing differences. In particular, the authors should do their best to identify likely reasons why the estimates in the previous studies have differed (if they do). For example, there are differences in the magnitude of the susceptibility (e.g., Wang et al., 2012 versus Terai et al., 2015). There are also differences in the behavior of susceptibility (monotonic decrease versus increase and then decrease).*

**We have provided more discussion on the differences in both magnitude and behavior of susceptibility in previous studies in the second paragraph of the Section 4 (Discussion). Now the text reads "Our results may help to reconcile some of the differences in previous estimates of precipitation susceptibility. For example, our results show that $S_{X\_AI} \approx 0.3 S_{X\_CDNC}$,(Table 3 and Fig. 1), which explains why $S_{POP\_CDNC}$ in Terai et al. (2015) is much larger than $S_{POP\_AI}$ in Wang et al., (2012). Previous studies are also different**

in how precipitation susceptibility varies with increasing LWP. Our results show that $S_I$ generally increases with LWP at low and moderate LWP and then decreases with increasing LWP at moderate and high LWP, consistent with results from Feingold et al., (2013), Michibata et al., (2016) and Jung et al., (2016). The monotonic increase of $S_{I\_CDNC}$ with increasing LWP in Terai et al., (2015) is mainly because that the LWP range in their study is relatively narrow (from 0 to ~400 g m$^{-2}$) and our results suggest that when the upper bound of LWP is extended to ~800 g m$^{-2}$, the "descending branch" (S decreases with increasing LWP) noted in Feingold et al. (2013) appears, though the exact LWP value where $S_{I\_CDNC}$ peaks depend on LWP and rain products used as well as the rainfall threshold choices."

*3. The authors seem to argue for the use of SPOP as a metric to quantify aerosol-cloud-precipitation interactions due to SPOP having a smaller range of possible values, based on different LWP and precipitation rate retrievals (Fig. 14). There is less discussion on the advantages and disadvantages of using CDNC or AI as a metric and also a lack of discussion on how the threshold (rain vs. drizzle) can significantly change SPOP values. Given that the authors have examined a wide range of potential sources that lead to differences in susceptibility estimates, it would be informative for the readers to have the authors synthesize their findings and discuss what should be considered in future attempts to try to observationally constrain precipitation susceptibility or attempts to compare susceptibilities from models and from observations.*

Thanks for your suggestions! We now made further recommendations on how to better use these metrics to quantify aerosol-cloud-precipitation interactions in models and observations in Section 5, and it reads:

"As $S_{POP}$ demonstrates relatively robust features across different LWP and rain products, this makes it a valuable metric for quantifying aerosol-cloud-precipitation interactions in observations and models. For instance, it would be highly interesting to examine why $S_{POP}$ strongly depends on atmospheric stability and how well this dependence is represented in a hierarchy of models (e.g., large eddy simulations, cloud resolving models, regional climate models, and global climate models). We also note that $S_{POP\_CDNC}$ is generally less uncertain compared to $S_{POP\_AI}$ and that a relatively robust relationship between $S_{POP\_CDNC}$ and $S_{POP\_AI}$ exists (i.e., $S_{X\_AI} \approx 0.3 S_{X\_CDNC}$) (Fig. 13 and Table 3). Given that aerosol retrievals near clouds are still challenging and aerosol-cloud relationships in satellite observations can be affected by aerosol retrieval contaminations from clouds, we recommend to first thoroughly quantify $S_{POP\_CDNC}$ in observations and models. As $S_{POP\_CDNC}$ is derived based on CDNC instead of AI, $S_{POP\_CDNC}$ is also not influenced by wet scavenging. Only after $S_{POP\_CDNC}$ is thoroughly quantified, we can then combine it with how CDNC depends on AI to better quantify $S_{POP\_AI}$.

On the other hand, $S_I$ estimates strongly depend on satellite retrieval products. Uncertainties in $S_I$ estimate are particular large when $S_I$ is estimated based on rain samples (> 0 dBZ) rather than drizzle samples (> -15 dBZ). It would then be desirable to use drizzle samples to estimate $S_I$. However, satellite retrieval of precipitation rate for drizzle can be highly uncertain. It is therefore recommended to further improve the retrieval accuracy of precipitation rate for drizzle in satellite observations in order to better use satellite estimate of $S_I$ to quantify aerosol-cloud precipitation interactions. Alternatively, long-term ground and in-situ observations with high accuracy precipitation rate retrievals can be used to provide better estimate $S_I$ and to further quantify aerosol-cloud-precipitation interactions.".

Further discussions are added on difference of $S_{POP}$ and $S_I$ between rain and drizzle in Section 5 and now the text reads " Our results suggest that onset of drizzle is not as readily suppressed by increases in AI or CDNC in warm clouds as rainfall (i.e., $S_{POP}$ is smaller for drizzle than for rain, especially at moderate LWP, Fig. 9). This may partly come from the fact that POP of drizzle is close to 100% at moderate and high LWP regardless of CDNC or AI values (Fig. 7a-d), which makes it insensitive to perturbations in CDNC or AI and results in smaller $S_{POP}$ at these LWP bins compared with $S_{POP}$ for rain (Fig. 9). On the other hand, precipitation intensity susceptibility is generally smaller for rain than for drizzle. This is consistent with our expectation that when precipitation intensity increases, accretion contributes more to the production of precipitation, which makes precipitation intensity less sensitive to perturbation in CDNC or AI, as accretion is less dependent on CDNC compared with autconversion (Feingold et al., 2013; Wood, 2005)"

*Minor Comments:*
*1. P2 L3: "Susceptibility is an inherent property of the aerosol-cloud system." – This is an interesting statement, but it is also vague. Does the statement mean that susceptibility doesn't change with cloud condition? Or aerosol condition? Should they be robust to differences in measurement platform, etc.?*
We agree this statement is indeed vague, and this statement is now removed in the revised manuscript.

*2. P5 L32: "... selected in close proximity of clouds pixels." What exact criteria is used to determine how close aerosol retrievals must be to be used in the study?*
Exact criteria for collocation are added in Sec 2.1. Now the text reads "…. MODIS cloud product and CPR radar reflectivity observations used in this study are both provided from the Caltrack datasets, which resample observations from many sensors under CALIOP subtrack with the horizontal resolution of 5km (see the website of

http://www.icare.univ-lille1.fr/projects/calxtract/products for more
information). For other aerosol and cloud products, including
MODIS/CALIOP aerosol products and AMSR-E cloud products, these are
further collocated into the CALIOP subtracks in the Caltrack dataset. For
each CALIOP subtrack, the closest aerosol and cloud retrieval samples
within one-degree grid box (1°×1°) centered at this subtrack are chosen. To
reduce the uncertainty in cloud retrievals, only samples where MODIS
cloud fraction is equal to 100% are selected".

*3. P6 L23: replace "significant" with "significantly"*
**Done.**

*4. P6 L25: What is the spatial resolution of the precipitation data? Is it at the
footprint level? In general, how are pairs of LWP, precipitation rate, and aerosol
proxy combined? Are they all combined at the footprint of the precipitation rate? Is
the coarsest footprint used for the comparison?*
**The horizontal resolution of all precipitation data used in the paper is at a
footprint level with 1.3km cross track and 1.7 km along track except CPR
radar reflectivity observations (i.e., 2B-GEOPROF product collocated to
CALIOP subtrack with 5km resolution). The resolution of different
products can be seen in Table2. Overall, MODIS LWP, precipitation rate
from 2B-GEOPROF and aerosol proxy are combined to CALIOP subtrack
since CALIOP aerosol product, MODIS cloud product and CPR 2B-GEOPROF
product used in the paper are all provided from Caltrack datasets. For
other retrieval products, including MODIS/CALIOP aerosol products and
AMSR-E cloud products, these are further collocated into the CALIOP
subtracks in the Caltrack dataset. For each CALIOP subtrack, the closest
aerosol and cloud retrieval samples within one-degree grid box (1°×1°) centered
at this subtrack are chosen. More details can be found in the first paragraph
of Sec 2.1 in the revised manuscript.**

*5. P7 L15: Provide some indication of statistical uncertainty in the estimates in Fig.
1. See main comment 1. Also, it would be informative to indicate the 0 value with a
dotted or gray line, because values below that line will indicate that increases in
aerosols/cloud droplets lead to more precipitation.*
**We have added error bars with 95% confidence intervals and zeroline in
the Fig. 1.**

*6. P7 L22-24: The turning point is very slight. The confidence intervals will be
helpful in determining how significant the peak is.*
**Thanks! We now added error bars. The peak is not significant anymore
after the error bars are added. But $S_{I\_CDNC}$ would decrease distinctly after
the peak if the upper bound of LWP and the number of LWP bins both
increased (see figure below). The sentence is now reformulated to**

"Although the $S_{I\_CDNC}$ peak (around 0.6 with LWP 350 gm$^{-2}$) is not significant in Fig. 1b, $S_{I\_CDNC}$ would decrease distinctly after the peak if the upper bound of LWP and the number of LWP bins both increase (not shown). This turning point may correspond to conversion process shifting from the autoconversion to accretion regime (Michibata et al., 2016)."

[Figure]

**Same as the Fig. 1b but with increase in the upper boundary of LWP and the number of LWP bins**

*7. P7 L29: To show that the fluctuations in the mean are small compared to noise, the interquartile range (between 25th percentile and 75th percentile) can be shown.*
**Done. The interquartile range is now added to Fig. 2.**

*8. P7 L29: Also, because the AI vs. CDNC relationship takes the form d ln(CDNC)/d ln(AI) and because it looks like the AI has a lognormal distribution, it might be better to plot the x-axis in log-scale.*
**Done. Now the x-axis in Fig. 2 is in log-scale.**

*9. P8 L4-5: The differences in dlnCDNC/dlnAI between the different stability regimes are interesting, in particular, the lack of sensitivity (or negative sensitivity at high LWPs). Are these differences significant? Do the authors have an explanation as to why the stability affects the sensitivity?*
**We now add error bars to Fig. 3, and now the differences in dlnCDNC/dlnAI between the different stability regimes and negative sensitivity at high LWP are both not significant anymore.**

*10. P8 L16: The subtle differences in Fig 4 are hard to see because of the large y-axis range. I can understand the choice to try to keep the same axis range across different figures, but in this case, I would suggest narrowing the range to allow the reader to discern any differences.*
**We have narrowed the range of y-axis and also added a zeroline to this figure in the revised manuscript.**

*11. P8 L26-28: Is there a reason why we would rely more heavily on and prefer MODIS AI rather than CALIPSO AI?*
**This is mainly because MODIS AI has been widely used in previous studies for examining aerosol-cloud-precipitation interactions. What is more, Costantino and Bréon, (2010) shown that AOD estimate from CALIPSO product was very noisy and less reliable than the equivalent parameter from MODIS. The 2D vs. 1D sampling is a likely reason for the MODIS AI being a bit smoother that the CALIPSO AI.**

*12. P9 L2-7: This is one case where confidence intervals can show that the SPOP estimates are not significantly affected by the choice of LWP retrievals, whereas the SI and SR estimate are significantly affected.*
**Thanks a lot for this excellent suggestion! After adding error bars to Fig.5, it indeed shows the discrepancies in $S_{POP}$ between MODIS and AMSR-E LWP are not significant. We have added this sentence to the first paragraph in Sec 3.3.**

*13. P9 L26: Data is plural, so it should be "... when data are binned..."*
**Corrected.**

*14. P10 L11: Although the axis labels show this, the figure caption to Figure 9 should indicate the difference between the top row and the bottom row.*
**We have clarified this in the caption of Fig. 9.**

*15. P10 L26-28: What is the impact on SR if SPOP increases and SI decreases with increases in the threshold?*
**$S_R$ is indeed not affected by the rainfall definition since mean rain rate for any given LWP/CDNC or LWP/AI bin is calculated for both rainy and non-rainy clouds, and does not depend on rainfall thresholds used to define a rain event. We have added this sentence to the third paragraph in Sec 3.4 and it reads "By contrast, $S_R$ is not affected by the rainfall definition since the mean rain rate R for a given LWP/CDNC or LWP/AI bin is calculated based on both rainy and non-rainy clouds and does not depend on rainfall thresholds (not shown)."**

*16. P10 L31: "more significant" should be replaced with "larger", because significant has a particular meaning in the literature (statistical significance), and to state more significant would require examining the confidence intervals.*
***Done.***

*17. P11 L28: "sigh" should be replaced by "sign"*
***Done.***

*18. P14 L5: Insert "by increases in AI or CDNC" between "readily suppressed" and*

*"in warm clouds"*
**Done.**

*19. P14 L5-6: Taken at face value, this statement is counterintuitive, isn't it? Wouldn't we expect rainfall, which is more dependent on accretion than on autoconversion, to have a weaker dependence to CDNC?*

**The above expectation is consistent with how $S_I$ changes with rainfall thresholds. When the rainfall threshold increases, it shifts the production of rain from autoconversion to accretion, which reduces precipitation intensity susceptibility. As for precipitation frequency susceptibility, it depends on how often precipitation frequency reaches its upper limit, 100%. As the rainfall threshold decreases from 0 dBZ to -15 dBZ, POP for drizzle is close to 100% at intermediate and high LWP as shown in the figure below, which make it insensitive to perturbation in CDNC or AI at intermediate and high LWP, resulting in much smaller $S_{POP}$ at these LWP bins as shown in Fig. 9 in the main text. We now added this discussion to the third paragraph in Sec.5. Now the text reads "Our results suggest that onset of drizzle is not as readily suppressed by increases in AI or CDNC in warm clouds as rainfall (i.e., $S_{POP}$ is smaller for drizzle than for rain, especially at moderate LWP, Fig. 9). This may partly come from the fact that POP of drizzle is close to 100% at moderate and high LWP regardless of CDNC or AI values (Fig. 7a-d), which makes it insensitive to perturbations in CDNC or AI and results in smaller $S_{POP}$ at moderate and high LWP bins compared with $S_{POP}$ for rain (Fig. 9). On the other hand, precipitation intensity susceptibility is generally smaller for rain than for drizzle. This is consistent with our expectation that when precipitation intensity increases, accretion contributes more to the production of precipitation, which makes precipitation intensity less sensitive to perturbation in CDNC or AI, as accretion is less dependent on CDNC compared with autconversion (Feingold et al., 2013; Wood, 2005). ".**

[Figure]

**Probability of precipitation as a function of MODIS LWP and its breakdown into drizzle (>0.14 mm d⁻¹) and rain (>2 mm d⁻¹)**

*20. P14 L14: Replace "value" with "values"*

**Done.**

*21. P14 L14-18: Are these results consistent with existing conceptual frameworks (such as those based on LES) on how stability affects aerosol-cloud-precipitation interactions? Are there LES studies that have addressed how stability might affect susceptibility?*

**The pattern of $S_{POP\_AI}$ under different stability conditions from our paper (Fig. 13b and Fig. 13f) is consistent with the findings of L'Ecuyer et al., (2009). In addition, Terai et al., (2015) found maximum $S_{POP\_CDNC}$ occurred in regions where stable regime is predominant. These satellite-based studies, however, did not provide physical interpretations of such results. Lebo and Feingold (2014) calculated precipitation susceptibility for stratocumulus and trade wind cumulus using large-eddy simulations(LES) and included an overview of precipitation susceptibility estimates in the ligature based on LES. However, their study focus on the relationship between precipitation susceptibility and cloud water response to aerosol perturbations, and did not examine how precipitation susceptibility might be different for clouds under different cloud regimes. We now added this discussion in the revised manuscript and calls further efforts to understand this difference, especially for $S_{POP}$ in the Section 5.**

**References:**

[revised manuscript text omitted]